# Progressive3D: Progressively Local Editing for Text-to-3D Content Creation with Complex Semantic Prompts

**Xinhua Cheng**[*1], **Tianyu Yang**[†2], **Jianan Wang**[2], **Yu Li**[2], **Lei Zhang**[2], **Jian Zhang**[1], **Li Yuan**[†1,3]
[1]Peking University
[2]International Digital Economy Academy (IDEA)
[3]Peng Cheng Laboratory

## Abstract

Recent text-to-3D generation methods achieve impressive 3D content creation capacity thanks to the advances in image diffusion models and optimizing strategies. However, current methods struggle to generate correct 3D content for a complex prompt in semantics, *i.e.*, a prompt describing multiple interacted objects binding with different attributes. In this work, we propose a general framework named **Progressive3D**, which decomposes the entire generation into a series of locally progressive editing steps to create precise 3D content for complex prompts, and we constrain the content change to only occur in regions determined by user-defined region prompts in each editing step. Furthermore, we propose an overlapped semantic component suppression technique to encourage the optimization process to focus more on the semantic differences between prompts. Experiments demonstrate that the proposed Progressive3D framework is effective in local editing and is general for different 3D representations, leading to precise 3D content production for prompts with complex semantics for various text-to-3D methods. Our project page is https://cxh0519.github.io/projects/Progressive3D/

## 1 Introduction

High-quality 3D digital content that conforms to the requirements of users is desired due to its various applications in the entertainment industry, mixed reality, and robotic simulation. Compared to the traditional 3D generating process which requests manual design in professional modeling software, automatically creating 3D content with given text prompts is more friendly for both beginners and experienced artists. Driven by the recent progress of neural 3D representations (Mildenhall et al., 2020; Wang et al., 2021; Yariv et al., 2021; Shen et al., 2021) and text-to-image (T2I) diffusion models (Nichol et al., 2021; Rombach et al., 2022; Mou et al., 2023; Zhang et al., 2023), Dreamfusion (Poole et al., 2022) demonstrates impressive 3D content creation capacity conditioned on given prompts by distilling the prior knowledge from T2I diffusion models into a Neural Radiance Field (NeRF), which attracts board interests and emerging attempts in text-to-3D creation.

Although text-to-3D methods have tried to use various 3D neural representations (Lin et al., 2023; Chen et al., 2023; Tsalicoglou et al., 2023) and optimization strategies (Wang et al., 2023a; Huang et al., 2023b; Wang et al., 2023b) for improving the quality of created 3D content and achieving remark accomplishments, they rarely pay attention to enhancing the semantic consistency between generated 3D content and given prompts. As a result, most text-to-3D methods struggle to produce correct results when the text prompt describes a complex scene involving multiple objects binding with different attributes. As shown in Fig. 1(a), existing text-to-3D methods suffer from challenges with complex prompts, leading to significant object missing, attribute mismatching, and quality reduction. While recent investigations (Feng et al., 2022; Huang et al., 2023a; Lu et al., 2023) have demonstrated that current T2I diffusion models tend to generate inaccurate results when facing

---

[*]Work done during the internship at IDEA.
[†]Corresponding Authors.

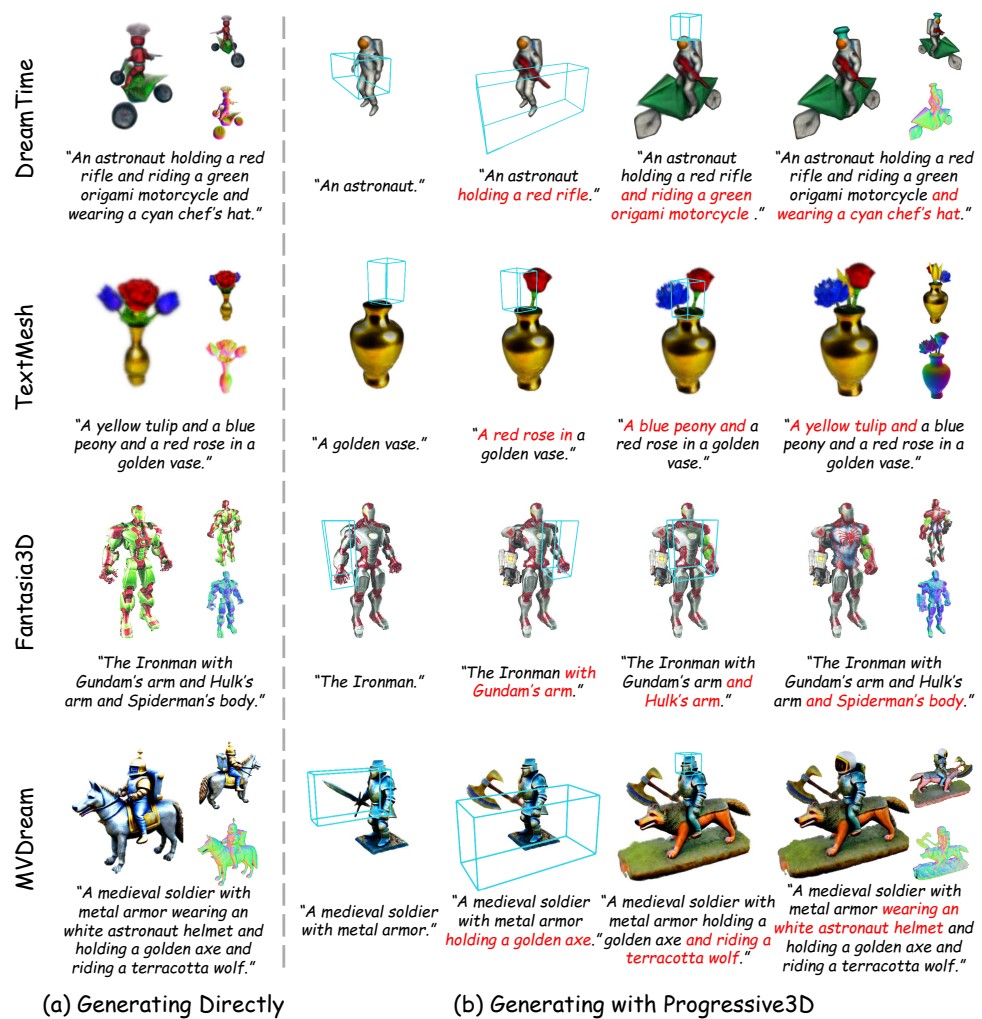

Figure 1: **Conception.** Current text-to-3D methods suffer from challenges when given prompts describing multiple objects binding with different attributes. Compared to (a) generating with existing methods, (b) generating with Progressive3D produces 3D content consistent with given prompts.

prompts with complex semantics and existing text-to-3D methods inherit the same issues from T2I diffusion models, works on evaluating or improving the performance of text-to-3D methods in complex semantic scenarios are still limited. Therefore, how to generate correct 3D content consistent with complex prompts is critical for many real applications of text-to-3D methods.

To address the challenges of generation precise 3D content from complex prompts, we propose a general framework named **Progressive3D**, which decomposes the difficult creation of complex prompts into a series of local editing steps, and progressively generates the 3D content as is shown in Fig. 1(b). For a specific editing step, our framework edits the pre-trained source representation in the 3D space determined by the user-defined region prompt according to the semantic difference between the source prompt and the target prompt. Concretely, we propose two content-related constraints, including a consistency constraint and an initialized constraint for keeping content beyond selected regions unchanged and promoting the separate target geometry generated from empty space. Furthermore, a technique dubbed Overlapped Semantic Component Suppression (OSCS) is carefully designed to automatically explore the semantic difference between the source prompt and the target one for guiding the optimization process of the target representations.

To evaluate Progressive3D, we construct a complex semantic prompt set dubbed CSP-100 consisting of 100 various prompts. Prompts in CSP-100 are divided into four categories including color, shape,

material and composition according to appeared attributes. Experiments conducted on existing text-to-3D methods driven by different 3D representations including NeRF-based DreamTime (Huang et al., 2023b) and MVDream(Shi et al., 2023), SDF-based TextMesh (Tsalicoglou et al., 2023), and DMTet-based Fantasia3D (Chen et al., 2023) demonstrate that our framework produces precise 3D models through multi-step local editing achieve better alignment with text prompts both in metrics and user studies than current text-to-3D creation methods when prompts are complex in semantics.

Our contribution can be summarized as follows: **(1)** We propose a framework named **Progressive3D** for creating precise 3D content prompted with complex semantics by decomposing a difficult generation process into a series of local editing steps. **(2)** We propose the Overlapped Semantic Component Suppression to sufficiently explore the semantic difference between source and target prompts for overcoming the issues caused by complex prompts. **(3)** Experiments demonstrate that Progressive3D is effective in local editing and is able to generate precise 3D content consistent with complex prompts with various text-to-3D methods driven by different 3D neural representations.

## 2 RELATED WORKS

**Text-to-3D Content Creation.** Creating high-fidelity 3D content from only text prompts has attracted broad interest in recent years and there are many earlier attempts (Jain et al., 2022; Michel et al., 2022; Mohammad Khalid et al., 2022). Driven by the emerging text-to-image diffusion models, Dreamfusion (Poole et al., 2022) firstly introduces the large-scale prior from diffusion models for 3D content creation by proposing the score distillation sampling and achieves impressive results. The following works can be roughly classified into two categories, many attempts such as SJC (Wang et al., 2023a), Latent-NeRF (Metzer et al., 2022), Score Debiasing (Hong et al., 2023) DreamTime (Huang et al., 2023b), ProlificDreamer (Wang et al., 2023b) and MVDream(Shi et al., 2023) modify optimizing strategies to create higher quality content, and other methods including Magic3D (Lin et al., 2023), Fantasia3D (Chen et al., 2023), and TextMesh (Tsalicoglou et al., 2023) employ different 3D representations for better content rendering and mesh extraction. However, most existing text-to-3D methods focus on promoting the quality of generated 3D content, thus their methods struggle to generate correct content for complex prompts since no specific techniques are designed for complex semantics. Therefore, we propose a general framework named Progressive3D for various neural 3D representations to tackle prompts with complex semantics by decomposing the difficult generation into a series of local editing processes, and our framework successfully produces precise 3D content consistent with the complex descriptions.

**Text-Guided Editing on 3D Content.** Compared to the rapid development of text-to-3D creation methods, the explorations of editing the generated 3D content by text prompts are still limited. Although Dreamfusion (Poole et al., 2022) and Magic3D (Lin et al., 2023) demonstrate that content editing can be achieved by fine-tuning existing 3D content with new prompts, such editing is unable to maintain 3D content beyond editable regions untouched since the fine-tuning is global to the entire space. Similar global editing methods also include Instruct NeRF2NeRF (Haque et al., 2023) and Instruct 3D-to-3D (Kamata et al., 2023), which extend a powerful 2D editing diffusion model named Instruct Pix2Pix (Brooks et al., 2023) into 3D content. Furthermore, several local editing methods including Vox-E (Sella et al., 2023) and DreamEditor (Zhuang et al., 2023) are proposed to edit the content in regions specified by the attention mechanism, and FocalDreamer (Li et al., 2023) only generates the incremental content in editable regions with new prompts to make sure the input content is unchanged. However, their works seldom consider the significant issues in 3D creations including object missing, attribute mismatching, and quality reduction caused by the prompts with complex semantics. Differing from their attempts, our Progressive3D emphasizes the semantic difference between source and target prompts, leading to more precise 3D content.

## 3 METHOD

Our Progressive3D framework is proposed for current text-to-3D methods to tackle prompts with complex semantics. Concretely, Progressive3D decomposes the 3D content creation process into a series of progressively local editing steps. For each local editing step, assuming we already have a source 3D representation $\phi_s$ supervised by the source prompt $\boldsymbol{y}_s$, we aim to obtain a target 3D representation $\phi_t$ which is initialized by $\phi_s$ to satisfy the description of the target prompt $\boldsymbol{y}_t$ and

the 3D region constraint of user-defined region prompts $y_b$. We first convert user-defined region prompts to 2D masks for each view separately to constrain the undesired contents in $\phi_t$ untouched (Sec. 3.1), which is critical for local editing. Furthermore, we propose the Overlapped Semantic Component Suppression (OSCS) technique to optimize the target 3D representation $\phi_t$ with the guidance of the semantic difference between the source prompt $y_s$ and the target prompt $y_t$ (Sec. 3.2) for emphasizing the editing object and corresponding attributes. The overview illustration of our framework is shown in Fig .2.

## 3.1 EDITABLE REGION DEFINITION AND RELATED CONSTRAINTS

In this section, we give the details of the editable region definition with a region prompt $y_b$ and designed region-related constraints. Instead of directly imposing constraints on neural 3D representations to maintain 3D content beyond selected regions unchanged, we adopt 2D masks rendered from 3D definitions as the bridge to connect various neural 3D representations (*e.g.*, NeRF, SDF, and DMTet) and region definition forms (*e.g.*, 3D bounding boxes, custom meshes, and 2D/3D segmentation results (Liu et al., 2023; Cheng et al., 2023; Cen et al., 2023)), which enhances the generalization of our Progressive3D. We here adopt NeRF as the neural 3D representation and define the editable region with 3D bounding box prompts for brevity.

Given a 3D bounding box prompt $y_b = [c_x, c_y, c_z; s_x, s_y, s_z]$ which is user-defined for specifying the editable region in 3D space, where $[c_x, c_y, c_z]$ is the coordinate position of the box center, and $[s_x, s_y, s_z]$ is the box size on the $\{x, y, z\}$-axis respectively. We aim to obtain the corresponding 2D mask $M_t$ converted from the prompt $y_b$ and pre-trained source representation $\phi_s$ that describes the editable region for a specific view $v$. Concretely, we first calculate the projected opacity map $\hat{O}$ and the projected depth map $\hat{D}$ of $\phi_s$ similar to the Eq. 10. Then we render the given bounding box to obtain its depth $D_b = render(y_b, v, R)$, where $v$ is the current view and $R$ is the rotate matrix of the bounding box. Before calculating the 2D editable mask $M_t$ at a specific $v$, we modify the projected depth map $\hat{D}$ according to $\hat{O}$ to ignore the floating artifacts mistakenly generated in $\phi_s$:

$$\tilde{D}(r) = \begin{cases} \infty, & \text{if } \hat{O}(r) < \tau_o; \\ \hat{D}(r), & \text{otherwise;} \end{cases} \tag{1}$$

where $r \in \mathcal{R}$ is the ray set of sampled pixels in the image rendered at view $v$, and $\tau_o$ is the filter threshold. Therefore, the 2D mask $M_t$ of the editable region, as well as the 2D opacity mask $M_o$, can be calculated for the following region-related constraints:

$$M_t(r) = \begin{cases} 1, & \text{if } D_b(r) < \tilde{D}(r); \\ 0, & \text{otherwise.} \end{cases} \quad M_o(r) = \begin{cases} 1, & \text{if } \hat{O}(r) > \tau_o; \\ 0, & \text{otherwise.} \end{cases} \tag{2}$$

**Content Consistency Constraint.** We emphasize that maintaining 3D content beyond user-defined editable regions unchanged during the training of the target representation $\phi_t$ is critical for 3D editing. We thus propose a content consistency constraint to impose the content between the target representation $\phi_t$ and the source representation $\phi_s$ to be consistent in undesired regions, which conditioned by our obtained 2D mask $M_t$ which represents the editable regions:

$$\mathcal{L}_{consist} = \sum_{r \in \mathcal{R}} \left( \bar{M}_t(r) M_o(r) \left\| \hat{C}_t(r) - \hat{C}_s(r) \right\|_2^2 + \bar{M}_t(r) \bar{M}_o(r) \left\| \hat{O}_t(r) \right\|_2^2 \right), \tag{3}$$

where $\bar{M}_t = 1 - M_t$ is the inverse editable mask, $\bar{M}_o = 1 - M_o$ is the inverse opacity mask, and $\hat{C}_s, \hat{C}_t$ are projected colors of $\phi_s, \phi_t$ respectively.

Instead of constraining the entire unchanged regions by color similarity, we divide such regions into a content region and an empty region according to the modified opacity mask $M_o$, and an additional term is proposed to impose the empty region remains blank during training. We separately constrain content and empty regions to avoid locking the backgrounds during the training, since trainable backgrounds are proved (Guo et al., 2023) beneficial for the quality of foreground generation.

**Content Initialization Constraint.** In our progressive editing steps, a usual situation is the corresponding 3D space defined by region prompts is empty. However, creating the target object from

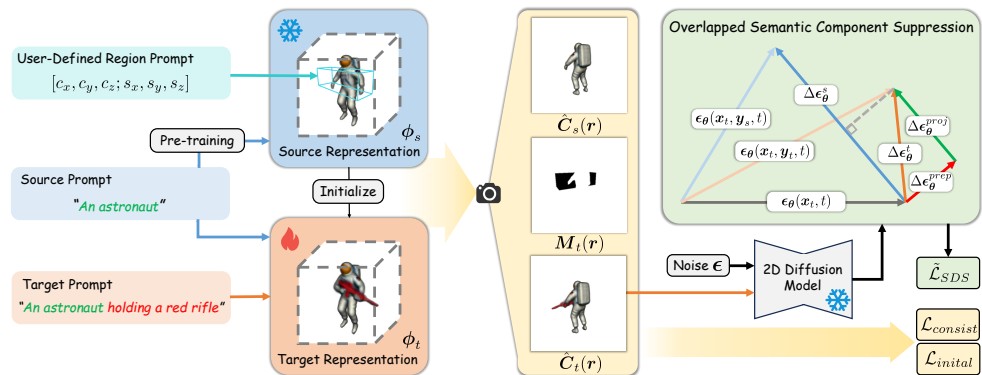

Figure 2: **Overview of a local editing step of our proposed Progressive3D.** Given a source representation $\phi_s$ supervised by source prompt $y_s$, our framework aims to generate a target representation $\phi_t$ conforming to the input target prompt $y_t$ in 3d space defined by the region prompt $y_b$. Conditioned on the 2D mask $M_t(r)$, we constrain the 3D content with $\mathcal{L}_{consist}$ and $\mathcal{L}_{inital}$. We further propose an Overlapped Semantic Component Suppression technique to impose the optimization focusing more on the semantic difference for precise progressive creation.

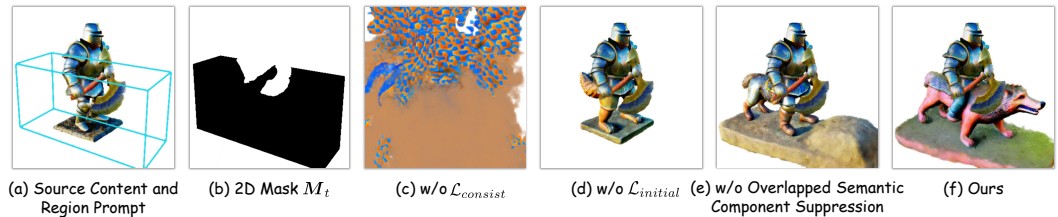

(a) Source Content and Region Prompt    (b) 2D Mask $M_t$    (c) w/o $\mathcal{L}_{consist}$    (d) w/o $\mathcal{L}_{inital}$    (e) w/o Overlapped Semantic Component Suppression    (f) Ours

Figure 3: **Qualitative ablations.** The source prompt $y_s$="*A medieval soldier with metal armor holding a golden axe.*" and the target prompt $y_t$="*A medieval soldier with metal armor holding a golden axe and riding a terracotta wolf.*", where green denotes the overlapped prompt and red denotes the different prompt.

scratch often leads to rapid geometry variation and causes difficulty in generation. We thus provide a content initialization constraint to encourage the user-defined 3D space filled with content, which is implemented by promoting $\hat{O}_t$ increase in editable regions during the early training phase:

$$\mathcal{L}_{inital} = \kappa(k) \sum_{r \in \mathcal{R}} M_t(r) \left\| \hat{O}_t(r) - \mathbf{1} \right\|_2^2; \quad \kappa(k) = \begin{cases} \lambda(1 - \dfrac{k}{K}), & \text{if } 0 \le k < K; \\ 0, & \text{otherwise,} \end{cases} \quad (4)$$

where $\kappa(k)$ is a weighting function of the current training iteration $k$, $\lambda$ is the scale factor of the maximum strength, and $K$ is the maximum iterations that apply this constraint to avoid impacting the detail generation in the later phase.

## 3.2 OVERLAPPED SEMANTIC COMPONENT SUPPRESSION

Although we ensure the content edits only occur in user-defined regions through region-related constraints, obtaining desired representation $\phi_t$ which matches the description in the target prompt $y_t$ is still challenging. An intuitive approach to create $\phi_t$ is fine-tuning the source representation $\phi_s$ with the target prompt $y_t$ directly (Poole et al., 2022; Lin et al., 2023). However, we point out that merely leveraging the target prompt $y_t$ for fine-grained editing will cause attribute mismatching issues, especially when $y_t$ describes multiple objects binding with different attributes.

For instance in Fig. 3, we have obtained a source representation $\phi_s$ matching the source prompt $y_s$ and a target prompt $y_t$ for the following local editing step. If we adjust $\phi_s$ guided by $y_t$ directly, as shown in Fig. 3(e), the additional content *"wolf"* could be both impacted by additional attribute *"terracotta"* and overlapped attribute *"metal, golden"* during the generation even if the overlapped attribute has been considered in $\phi_s$, which leads to an undesired result with attribute confusing.

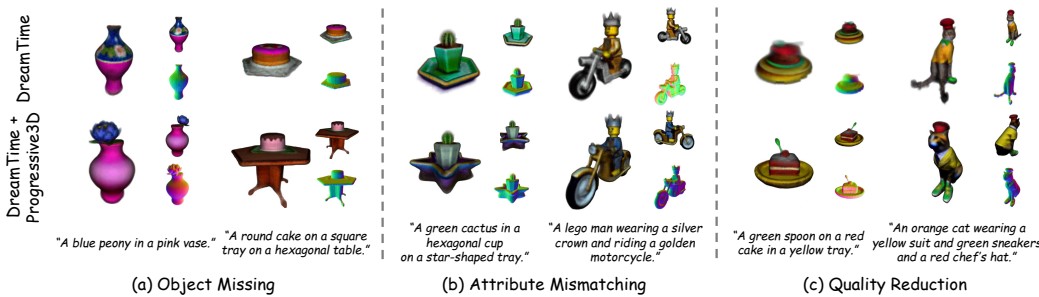

Figure 4: Current text-to-3D methods often fail to produce precise results when the given prompt describes multiple interacted objects binding with different attributes, leading to significant issues including object missing, attribute mismatching, and quality reduction.

Furthermore, the repeated attention to overlapped prompts causes the editing process less consider the objects described in additional prompts, leading to entire or partial object ignoring (*e.g.,* "*wolf*" is mistakenly created without its head and integrated with the soldier). Hence, guiding the optimization in local editing steps to focus more on the semantic difference between $y_s$ and $y_t$ instead of $y_t$ itself is critical for alleviating attribute mismatching and obtaining desired 3D content.

Therefore, we proposed a technique named Overlapped Semantic Component Suppression (OSCS) inspired by (Armandpour et al., 2023) to automatically discover the overlapped semantic component between $y_s$ and $y_t$ with vector projection, and OSCS then suppresses the overlapped component to enhance the influence of the different semantic during the training of $\phi_t$ for precise content creation. Concretely, both prompts $y_s$ and $y_t$ firstly produce separate denoising components with the unconditional prediction $\epsilon_{\boldsymbol{\theta}}(\boldsymbol{x}_t, t)$:

$$\Delta \epsilon_{\boldsymbol{\theta}}^s = \epsilon_{\boldsymbol{\theta}}(\boldsymbol{x}_t, \boldsymbol{y}_s, t) - \epsilon_{\boldsymbol{\theta}}(\boldsymbol{x}_t, t); \quad \Delta \epsilon_{\boldsymbol{\theta}}^t = \epsilon_{\boldsymbol{\theta}}(\boldsymbol{x}_t, \boldsymbol{y}_t, t) - \epsilon_{\boldsymbol{\theta}}(\boldsymbol{x}_t, t). \tag{5}$$

As shown in Fig. 2, we then decompose $\Delta \epsilon_{\boldsymbol{\theta}}^t$ into the projection component $\Delta \epsilon_{\boldsymbol{\theta}}^{proj}$ and the perpendicular component $\Delta \epsilon_{\boldsymbol{\theta}}^{prep}$ by projecting $\Delta \epsilon_{\boldsymbol{\theta}}^t$ on $\Delta \epsilon_{\boldsymbol{\theta}}^s$:

$$\Delta \epsilon_{\boldsymbol{\theta}}^t = \underbrace{\frac{\langle \Delta \epsilon_{\boldsymbol{\theta}}^s, \Delta \epsilon_{\boldsymbol{\theta}}^t \rangle}{||\Delta \epsilon_{\boldsymbol{\theta}}^s||^2} \Delta \epsilon_{\boldsymbol{\theta}}^s}_{\text{Projection Component}} + \underbrace{\left( \Delta \epsilon_{\boldsymbol{\theta}}^t - \frac{\langle \Delta \epsilon_{\boldsymbol{\theta}}^s, \Delta \epsilon_{\boldsymbol{\theta}}^t \rangle}{||\Delta \epsilon_{\boldsymbol{\theta}}^s||^2} \Delta \epsilon_{\boldsymbol{\theta}}^s \right)}_{\text{Perpendicular Component}} = \Delta \epsilon_{\boldsymbol{\theta}}^{proj} + \Delta \epsilon_{\boldsymbol{\theta}}^{prep}, \tag{6}$$

where $\langle \cdot, \cdot \rangle$ denotes the inner product. We define $\Delta \epsilon_{\boldsymbol{\theta}}^{proj}$ as the overlapped semantic component since it is the most correlated component from $\Delta \epsilon_{\boldsymbol{\theta}}^t$ to $\Delta \epsilon_{\boldsymbol{\theta}}^s$, and regard $\Delta \epsilon_{\boldsymbol{\theta}}^{prep}$ as the different semantic component which represents the most significant difference in semantic direction. Furthermore, we suppress the overlapped semantic component $\Delta \epsilon_{\boldsymbol{\theta}}^{proj}$ during training for reducing the influence of appeared attributes, and the noise sampler with OSCS is formulated as:

$$\hat{\epsilon}_{\boldsymbol{\theta}}(\boldsymbol{x}_t, \boldsymbol{y}_s, \boldsymbol{y}_t, t) = \epsilon_{\boldsymbol{\theta}}(\boldsymbol{x}_t, t) + \frac{\omega}{W} \Delta \epsilon_{\boldsymbol{\theta}}^{proj} + \omega \Delta \epsilon_{\boldsymbol{\theta}}^{prep}; \quad W > 1, \tag{7}$$

where $\omega$ is the original guidance scale in CFG described in Eq. 14, and $W$ is the weight to control the suppression strength for the overlapped semantics. We highlight that $W > 1$ is important for the suppression, since $\hat{\epsilon}_{\boldsymbol{\theta}}(\boldsymbol{x}_t, \boldsymbol{y}_s, \boldsymbol{y}_t, t)$ is degenerated to $\hat{\epsilon}_{\boldsymbol{\theta}}(\boldsymbol{x}_t, \boldsymbol{y}_t, t)$ when $W = 1$. Therefore, the modified Score Distillation Sampling (SDS) with OSCS is formulated as follows:

$$\nabla_{\boldsymbol{\phi}} \tilde{\mathcal{L}}_{\text{SDS}}(\boldsymbol{\theta}, \boldsymbol{x}) = \mathbb{E}_{t, \epsilon} \left[ w(t) (\hat{\epsilon}_{\boldsymbol{\theta}}(\boldsymbol{x}_t, \boldsymbol{y}_s, \boldsymbol{y}_t, t) - \epsilon) \frac{\partial \boldsymbol{x}}{\partial \boldsymbol{\phi}} \right]. \tag{8}$$

Compared to Fig. 3(e), leveraging OSCS effectively reduces the distraction of appeared attributes and assists Progressive3D in producing desired 3D content, as is shown in Fig. 3(f).

## 4 EXPERIMENTS

### 4.1 EXPERIMENTAL SETTINGS

We only provide important experimental settings including dataset, metrics, and baselines here due to the page limitation, more detailed experimental settings can be found at Appendix B.

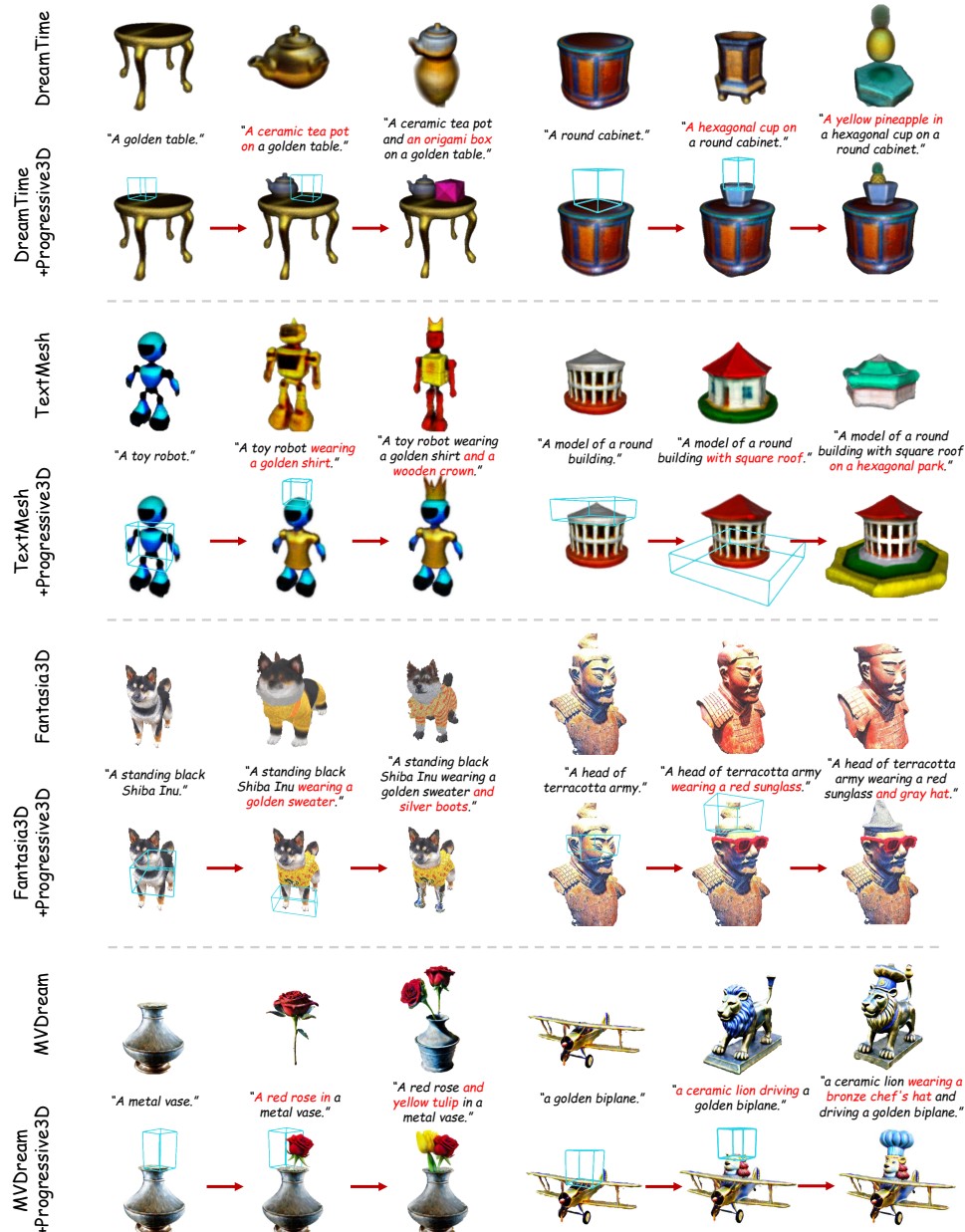

Figure 5: Progressive editing processes driven by various text-to-3D methods equipped with our Progressive3D. Compared to original methods, Progressive3D assists current methods in tackling prompts with complex semantics well. 3D Cyan boxes denote the user-defined region prompts.

**Dataset Construction.** We construct a Complex Semantic Prompt set named CSP-100 which involves 100 complex prompts to verify that current text-to-3D methods suffer issues when prompts are complex in semantics and proposed Progressive3D efficiently alleviates these issues. CSP-100 introduces four sub-categories of prompts including color, shape, material, and composition according to the appeared attribute and more details are in Appendix B.

**Evaluation Metrics.** Existing text-to-3D methods (Poole et al., 2022; Tsalicoglou et al., 2023; Li et al., 2023) leverage CLIP-based metrics to evaluate the semantic consistency between generated 3D creations and corresponding text prompts. However, CLIP-based metrics are verified (Huang et al., 2023a; Lu et al., 2023) that fail to measure the fine-grained correspondences between described objects and binding attributes. We thus adopt two recently proposed metrics fine-grained

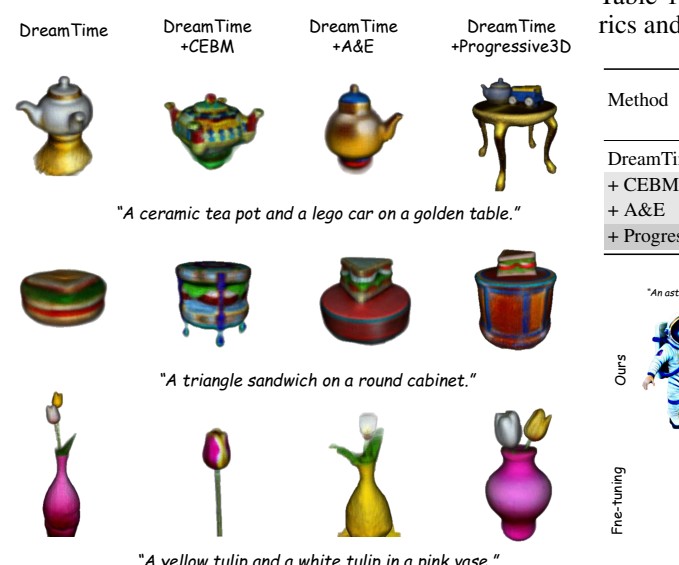

Table 1: Quantitative comparison on metrics and user studies over CSP-100.

| Method | Metrics | | Human |
|---|---|---|---|
| | B-VQA ↑ | mGPT-CoT ↑ | Preference ↑ |
| DreamTime | 0.227 | 0.522 | 16.8% |
| + CEBM | 0.186 | 0.491 | - |
| + A&E | 0.243 | 0.528 | - |
| + Progressive3D | **0.474** | **0.609** | **83.2%** |

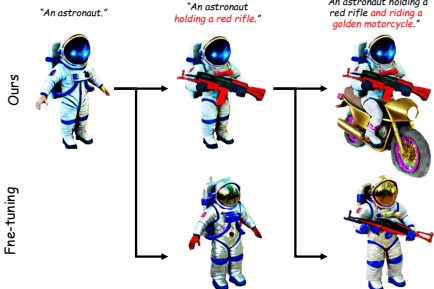

Figure 6: Visual comparison with DreamTime-based compositional generation baselines.

Figure 7: Qualitative ablations between fine-tuning with target prompts and editing with Progressive3D on MVDream.

including BLIP-VQA and mGPT-CoT (Huang et al., 2023a), evaluate the generation capacity of current methods and our Progressive3D when handling prompts with complex semantics.

**Baselines.** We incorporate our Progressive3D with 4 text-to-3D methods driven by different 3D representations: **(1)** DreamTime (Huang et al., 2023b) is a NeRF-based method which enhances DreamFusion (Poole et al., 2022) in time sampling strategy and produce better results. We adopt DreamTime as the main baseline for quantitative comparisons and ablations due to its stability and training efficiency. **(2)** TextMesh (Tsalicoglou et al., 2023) leverages SDF as the 3D representation to improve the 3D mesh extraction capacity. **(3)** Fantasia3D (Tsalicoglou et al., 2023) is driven by DMTet which produces impressive 3D content with a disentangled modeling process. **(4)** MV-Dream (Shi et al., 2023) is a NeRF-based method which leverages a pre-trained multi-view consistent text-to-image model for text-to-3D generation and achieves high-quality 3D content generation performance. To further demonstrate the effectiveness of Prgressive3D, we re-implement two composing text-to-image methods including Composing Energy-Based Model (CEBM) (Liu et al., 2022) and Attend-and-Excite (A&E) (Chefer et al., 2023) on DreamTime for quantitative comparison.

### 4.2 PROGRESSIVE3D FOR TEXT-TO-3D CREATION AND EDITING

**Comparison with current methods.** We demonstrate the superior performance of our Progressive3D compared to current text-to-3D methods in both qualitative and quantitative aspects in this section. We first present visualization results in Fig. 4 to verify that DreamTime faces significant challenges including (a) object missing, (b) attribute mismatching, and (c) quality reduction when given prompts describe multiple interacted objects binding with different attributes. Thanks to our careful designs, Progressive3D effectively promotes the creation performance of DreamTime when dealing with complex prompts. In addition, more progressive editing processes based on various text-to-3D methods driven by different neural 3D representations are shown in Fig. 5, which further demonstrate that Progressive3D stably increases the generation capacity of based methods when given prompts are complex, and our framework is general for various current text-to-3D methods.

We also provide quantitative comparisons on fine-grained semantic consistency metrics including BLIP-VQA and mGPT-CoT, and the results are shown in Tab. 1, which verify that our Progressive3D achieves remarkable improvements for 3D content creation with complex semantics compared to DreamTime-based baselines. As shown in Fig. 6, baselines that combine 2D composing T2I methods including CEBM (Liu et al., 2022) and A&E (Chefer et al., 2023) with DreamTime

Table 2: Quantitative ablation studies for proposed constraints and the OSCS technique based on DreamTime over CSP-100.

| Index | Components | | | Metrics | |
|---|---|---|---|---|---|
| | $\mathcal{L}_{consist}$ | $\mathcal{L}_{initial}$ | OSCS | B-VQA ↑ | mGPT-CoT ↑ |
| 1 | ✓ | | | 0.255 | 0.567 |
| 2 | ✓ | ✓ | | 0.370 | 0.577 |
| 3 | ✓ | | ✓ | 0.347 | 0.581 |
| 4 | ✓ | ✓ | ✓ | **0.474** | **0.609** |

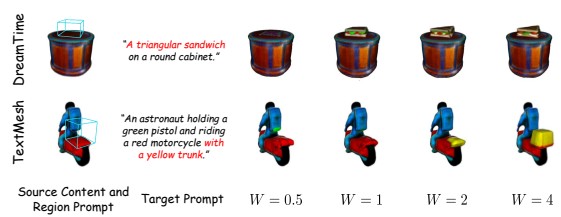

Figure 8: Qualitative ablations for suppression weight $W$ in proposed OSCS.

still achieve limited performance for complex 3D content generation, leading to significant issues including object missing, attribute mismatching, and quality reduction. Furthermore, we collected 20 feedbacks from humans to investigate the performance of our framework. The human preference shows that users prefer our Progressive3D in most scenarios (16.8% *vs.* 83.2%), demonstrating that our framework effectively promotes the precise creation capacity of DreamTime when facing complex prompts.

## 4.3 ABLATION STUDIES

In this section, we conduct ablation studies on DreamTime and TextMesh to demonstrate the effectiveness of proposed components including content consistency constraint $\mathcal{L}_{consist}$, content initialization constraint $\mathcal{L}_{initial}$ and Overlapped Semantic Component Suppression (OSCS) technique, we highlight that a brief qualitative ablation is given in Fig. 3.

We first present ablation results between fine-tuning directly and editing with Progressive3D based on TextMesh in Fig. 7 to demonstrate that fine-tuning with new prompts cannot maintain source objects prompted by overlapped semantics untouched and is unusable for progressive editing. Another visual result in Fig. 8 shows the parameter analysis of the suppression weight $w$ in OSCS. With the increase of $W$ (*i.e.*, $\frac{\omega}{W}$ decreases), the different semantics between source and target prompts play more important roles in optimizations and result in more desirable 3D content. On the contrary, the progressive step edits failed results with object missing or attribute mismatching issues when we increase the influence of overlapped semantics by setting $W = 0.5$, which further proves that our explanation of perpendicular and projection components is reasonable.

We then show the quantitative comparison in Tab. 2 to demonstrate the effectiveness of each proposed component, where content consistency constraint is not involved in quantitative ablations since consistency is the foundation of 3D content local editing which guarantees content beyond user-defined regions untouched. We underline that $\mathcal{L}_{initial}$ is proposed to simplify the geometry generation from empty space and OSCS is designed to alleviate the distraction of overlapped attributes, thus both components can benefit the creation performance with no conflict theoretically. This has been proofed by the quantitative ablations in Tab. 2: index 2 and 3 show that applying $\mathcal{L}_{initial}$ and OSCS alone both promote the metrics compared to the baseline in index 1, and index 4 shows that leveraging both $\mathcal{L}_{initial}$ and OSCS together can further contribute to the creation performance over CSP-100.

## 5 CONCLUSION

In this work, we propose a general framework named Progressive3D for correctly generating 3D content when the given prompt is complex in semantics. Progressive3D decomposes the difficult creation process into a series of local editing steps and progressively generates the aiming object with binding attributes with the assistance of proposed region-related constraints and the overlapped semantic suppression technique in each step. Experiments conducted on complex prompts in CSP-100 demonstrate that current text-to-3D methods suffer issues including object missing, attribute mismatching, and quality reduction when given prompts are complex in semantics, and the proposed Progressive3D effectively creates precise 3D content consistent with complex prompts through multi-step local editing. More discussions on the limitations and potential directions for future works are provided in Appendix A.

ACKNOWLEDGMENTS

This work was supported by the National Key R&D Program of China (2022ZD0118101), the Natural Science Foundation of China (No.62202014), Shenzhen Basic Research Program under Grant JCYJ20220813151736001, and also sponsored by CCF Tencent Open Research Fund.

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

## A    DISCUSSIONS

### A.1    REALISTIC USAGE OF PROGRESSIVE3D

We note that Progressive3D contains multiple local editing steps for creating complex 3D content, which accords with user usage pipeline, *i.e.,* creating a primary object first, then adjusting its attribute or adding more related objects. However, Progressive3D is flexible in realistic usage since the generation capacity of basic text-to-3D method and user goals are variant. For instance in Fig. 9, we desire to create the 3D content consistent with the prompt "An astronaut wearing a green top hat and riding a red horse". We find that MVDream fails to create the precise result while generating "An astronaut riding a red horse" correctly. Thus the desired content can be achieved by editing "An astronaut riding a red horse" within one-step editing, instead of starting from "an astronaut".

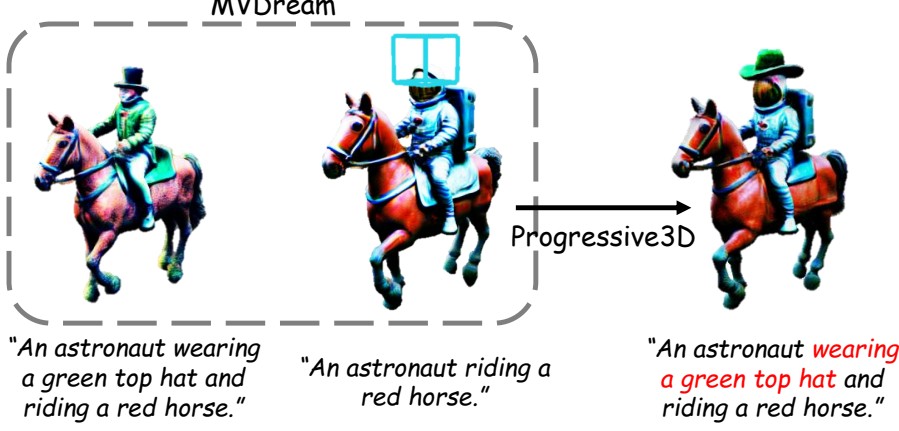

Figure 9: MVDream successfully creates "An astronaut riding a red horse" while failing to create "An astronaut wearing a green top hat and riding a red horse" By leveraging one-step Progressive3D editing, correct 3D content is obtained.

### A.2    OBJECT GENERATING ORDER

Different object generating orders in Progressive3D typically result in correct 3D content consistent with the complex prompts. However, the content details of the final content are impacted by created objects since Progressive3D is a local editing chain started from the source content, and we give an instance in Fig. 10. With different generating orders, Progressive3D creates 3D content with different details while they are both consistent with the prompt "An astronaut sitting on a wooden chair". In our experiments, we first generate the primary object which is desired to occupy most of the space and interact with other additional objects.

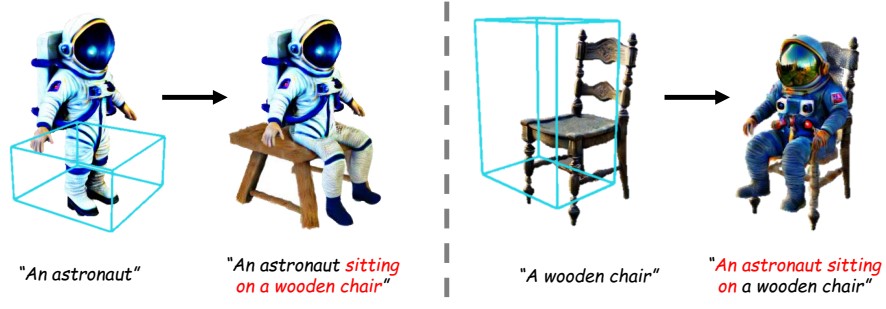

Figure 10: Creating "An astronaut sitting on a wooden chair" from different generating orders.

### A.3 Various Region Definitions

We highlight that Progressive3D is a general framework for various region definition forms since the corresponding 2D mask of each view can be achieved. As shown in Fig. 11, Progressive3D performs correctly on various definition forms including multiple 3D bounding boxes, custom mesh, and the fine-grained 2D segmentation prompted by the keyword "*helmet*" through Grounded-SAM (Liu et al., 2023; Kirillov et al., 2023).

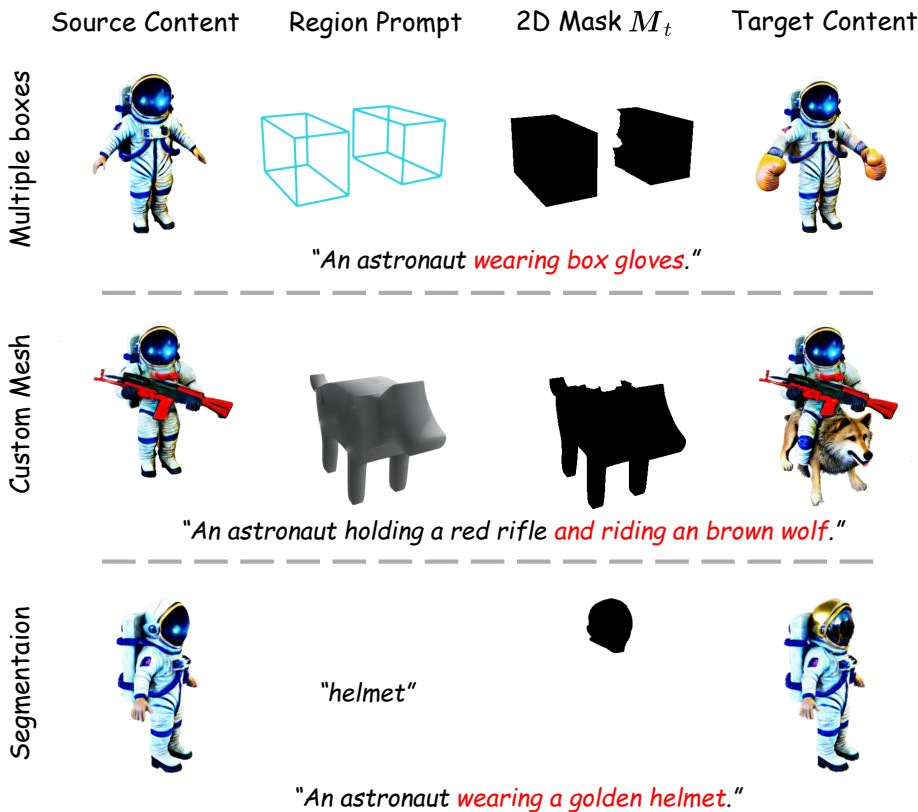

Figure 11: Progressive3D supports various definition forms of regions since the corresponding 2D masks of each view can be obtained.

### A.4 Attribute Editing

We emphasize that Progressive3D supports both modifying attributes of existing objects and creating additional objects with attributes not mentioned in source prompts from scratch in user-selected regions, and we provide attribute editing results in Fig. 12. Noticing that creating additional objects with attributes not mentioned in source prompts from scratch is more difficult than editing the attributes of existing objects. Therefore, attribute editing costs significantly less time than additional object generation.

### A.5 Why Adopting 2D Constraints

Compared to directly maintaining 3D content beyond editable regions unchanged on 3D representations, we adopt 2D constraints for achieving such a goal from the perspective of generalization.

We notice that current text-to-3D methods are developed based on various neural 3D representations, thus most 3D editing methods propose careful designs for a specific representation and are unusable on other representations. For instance, DreamEditor (Zhuang et al., 2023) distills the original NeRF into a mesh-based radiance field to localize the editable regions, and FocalDreamer (Li

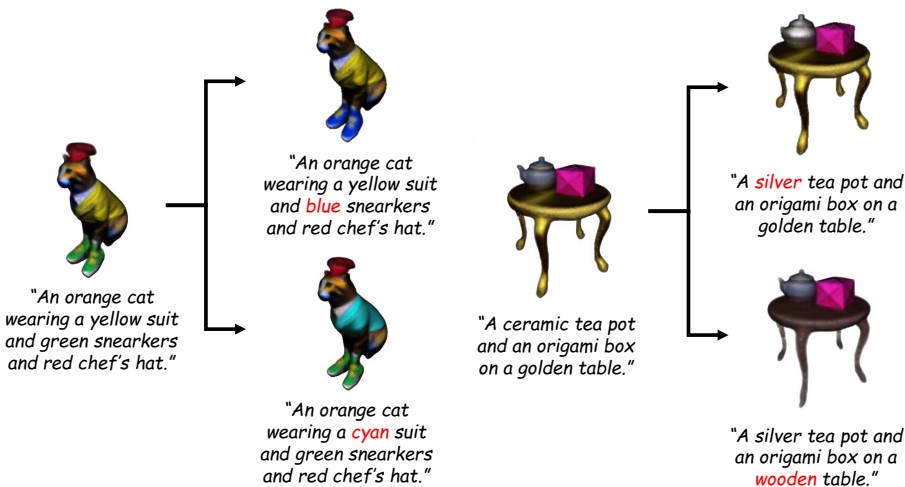

Figure 12: Prgressive3D supports attribute editing on existing generated objects.

et al., 2023) proposes multiple regularized losses specifically designed for DMTet to maintain the undesired regions unchanged. In addition, their specific designs also limit the available definition forms of editable regions.

However, we underline that the optimization core of most text-to-3D is the SDS loss which is supervised on rendered views in a 2D way, and different neural 3D representations can be rendered as 2D projected color, opacity, and depth maps through volume rendering or rasterization. Therefore, our proposed 2D region-related constraints can effectively bridge different representations and various user-provided definition forms, and the strength weights of our region-related constraints are easy to adjust since our constraints and SDS loss are all imposed on 2D views.

### A.6 LIMITATIONS

Our Progressive3D efficiently promotes the generation capacity for current text-to-3d methods when facing complex prompts. However, Progressive3D still faces several challenges.

Firstly, Progressive3D decomposes a difficult generation into a series of editing processes, which leads to multiplying time costs and more human intervention. A potential future direction is further introducing layout generation into the text-to-3d area, *e.g.*, creating 3D content with complex semantics in one generation by inputting a global prompt, a string of pre-defined 3D regions, and their corresponding local prompts. Whereas 3D layout generation intuitive suffers more difficulties and requires further exploration.

Another problem is that the creation quality of Progressive3D is highly determined by the generative capacity of the base method. We believe our framework can achieve better results when stronger 2D text-to-image diffusion models and neural 3D representations are available, and we leave the adaption between possible improved text-to-3D creation methods and Progressive3D in future works.

## B EXPERIMENTS SETTINGS

### B.1 CSP-100

CSP-100 can be divided into four sub-categories of prompts including color (*e.g.* red, green, blue), shape (*e.g.* round, square, hexagonal), material (*e.g.* golden, wooden, origami) and composition according to the attribute types appeared in prompts, as shown in Fig. 13. Each prompt in the color/shape/material categories describes two objects binding with color/shape/material attributes, and each prompt in the composition category describes at least three interacted objects with corresponding different attributes. We provide the detailed prompt list in both Tab. 4 and Tab. 5.

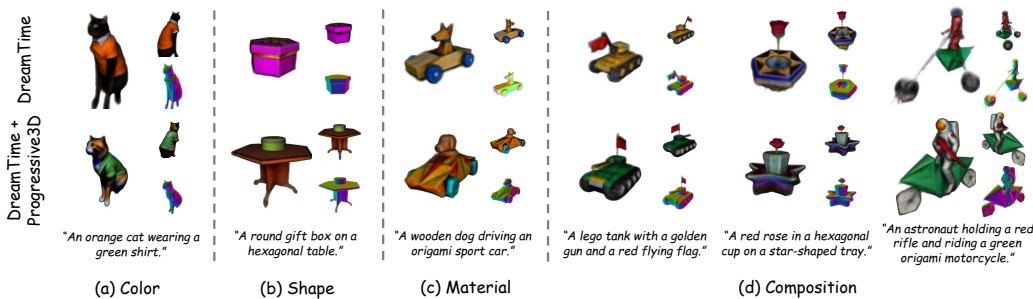

Figure 13: Prompts in CSP-100 can be divided into four categories including Color, Shape, Material, and Composition according to appeared attributes.

## B.2 METRICS

The CLIP-based metrics utilized by current text-to-3D methods (Poole et al., 2022; Tsalicoglou et al., 2023; Li et al., 2023) calculates the cosine similarity between text and image features extracted by CLIP (Radford et al., 2021) However, recent works (Huang et al., 2023a; Lu et al., 2023) demonstrate that CLIP-based metrics can only measure the coarse text-image similarity but fail to measure the fine-grained correspondences among multiple objects and their binding attributes. Therefore, we adopt two fine-grained text-to-image evaluation metrics including BLIP-VQA and mGPT-CoT proposed by (Huang et al., 2023a) to show the effectiveness of Progressive3D. We provide comparisons in Fig. 14 to demonstrate that the CLIP metric fails to measure the fine-grained correspondences while BLIP-VQA performs well, and we report the quantitative comparison of DreamTime-based methods on CLIP metric over CSP-100 in Tab. 3.

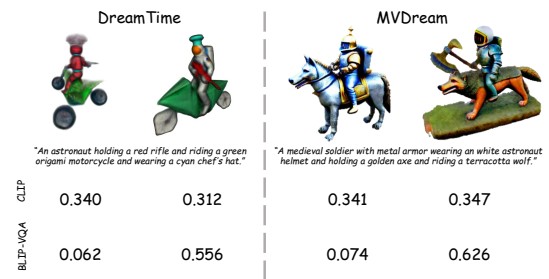

Figure 14: Quantitative comparison for metrics including CLIP and BLIP-VQA on DreamTime and MVDream.

Table 3: Quantitative comparison on CLIP over CSP-100.

| Method | CLIP ↑ |
|---|---|
| DreamTime | 0.289 |
| + CEBM | 0.275 |
| + A&E | 0.281 |
| + Progressive3D | **0.292** |

BLIP-VQA is proposed based on the visual question answering (VQA) ability of BLIP (Li et al., 2022). Concretely, BLIP-VQA decomposes a complex prompt into several separate questions and takes the probability of answering *"yes"* as the score for a question. The final score of a specific prompt is the product of the probability of answering *"yes"* for corresponding questions. For instance, the complex prompt is *"An astronaut holding a red rifle."*, the final score of BLIP-VQA is the product of the probability for questions including *"An astronaut?"* and *"A red rifle?"*

Since multimodal large language models such as MiniGPT-4 (Zhu et al., 2023) show impressive text-image understanding capacity, MiniGPT4 with Chain-of-Thought (mGPT-CoT) is proposed to leverage such cross-modal understanding performance to evaluate the fine-grained semantic similarity between query text and image. Specifically, we ask two questions in sequence including *"Describe the image."* and *"Predict the image-text alignment score."*, and the multimodal LLM is required to output the evaluation mGPT-CoT score with detailed Chain-of-Thought prompts. In practice, we adopt MiniGPT4 fine-tuned from Vicuna 7B (Chiang et al., 2023) as the LLM.

What's more, we define the preference criterion of human feedback as follows: Users are requested to judge whether the 3D creations generated by DreamTime or Progressive3D are consistent with the given prompts. If one 3D content is acceptable in the semantic aspect while the other is not, the corresponding acceptable method is considered superior. On the contrary, if both 3D creations are

considered to satisfy the description in the given prompt, users are asked to prefer the 3D content with higher quality.

### B.3 IMPLEMENT DETAILS

Our Progressive3D is implemented based on the Threestudio (Guo et al., 2023) project since Dream-Time (Huang et al., 2023b) and TextMesh (Tsalicoglou et al., 2023) have not yet released their source code and the official Fantasia3D (Chen et al., 2023) code project, and all experiments are conducted on NVIDIA A100 GPUs. We underline that the implementation in ThreeStudio (Guo et al., 2023) and details could be different from their papers, especially ThreeStudio utilizes Deep-Floyd IF (DeepFloyd-Team, 2023) as the text-to-image diffusion model for more stable and higher quality in the generation, while Fantasia3D adopts Stable Diffusion (Rombach et al., 2022) as the 2D prior. The number of iterations $N$ and batch size for DreamTime and TextMesh are set to 10000 and 1 respectively. The training settings of Fantasia3D are consistent with officially provided training configurations, *e.g.*, $N$ is 3000 and 2000 for geometry and appearance modeling stages, and batch size is set to 12. We leverage Adam optimizer for progressive optimization and the learning rate is consistent with base methods. Therefore, one local editing step costs a similar time to one generation of base methods from scratch. In most scenarios, the filter threshold $\tau_o$ is set to 0.5, the strength factor $\lambda$, iteration threshold $K$ in $\mathcal{L}_{consist}$ are set to 0.5 and $\frac{N}{4}$, and the suppression weight $W$ in overlapped semantic component suppression technique is set to 4.

## C QUALITATIVE RESULTS

### C.1 CONTENT CONSTRAINT WITH BACKGROUND

We divide the $\mathcal{L}_{consist}$ into a content term and an empty term to avoid mistakenly treating backgrounds as a part of foreground objects. We give a visual comparison of restricting the backgrounds and foregrounds as an entirety in Fig. 15, and the $\hat{\mathcal{L}}_{consist}$ can be formulated as:

$$\hat{\mathcal{L}}_{consist} = \sum_{\boldsymbol{r} \in \mathcal{R}} \left( \bar{M}_t(\boldsymbol{r}) \left\| \hat{\boldsymbol{C}}_t(\boldsymbol{r}) - \hat{\boldsymbol{C}}_s(\boldsymbol{r}) \right\|_2^2 \right). \tag{9}$$

The visual results demonstrate that mistakenly treating backgrounds as a part of foreground objects leads to significant floating.

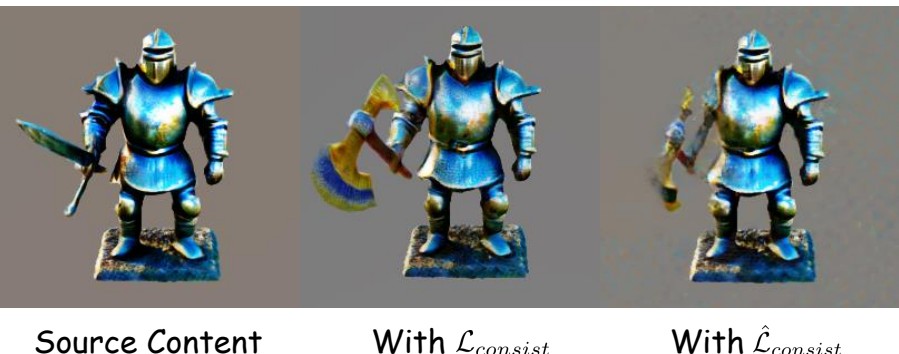

Source Content            With $\mathcal{L}_{consist}$            With $\hat{\mathcal{L}}_{consist}$

Figure 15: Mistakenly restricting background content as foreground leads to significant floating in editing result.

### C.2 MORE PROGRESSIVE EDITING PROCESS

We here provide more progressive editing results for correctly creating 3D content for prompts with complex semantics in Fig. 16. More qualitative results with various complex prompts further demonstrate the creation precision and diversity of our Progressive3D.

# D PRELIMINARY

**Neural Radiance Field** (NeRF) (Mildenhall et al., 2020) uses a multi-layer perception (MLP) to implicitly represent the 3D scene as a continuous volumetric radiance field. Specifically, MLP $\boldsymbol{\theta}$ maps a spatial coordinate and a view direction to a view-independent density $\sigma$ and view-dependent color $\boldsymbol{c}$. Given the camera ray $\boldsymbol{r}(k) = \boldsymbol{o} + k\boldsymbol{d}$ with camera position $\boldsymbol{o}$, view direction $\boldsymbol{d}$ and depth $k \in [k_n, k_f]$, the projected color of $\boldsymbol{r}(k)$ is rendered by sampling $N$ points along the ray:

$$\hat{\boldsymbol{C}}(\boldsymbol{r}) = \sum_{i=1}^{N} \Omega_i (1 - \exp(-\rho_i \delta_i))\boldsymbol{c}_i, \tag{10}$$

where $\rho_i$ and $\boldsymbol{c}_i$ denote the density and color of $i$-th sampled point, $\Omega_i = \exp(-\sum_{j=1}^{i-1} \rho_j \delta_j)$ indicates the accumulated transmittance along the ray, and $\delta_i$ is the distance between adjacent points.

**Diffusion Model** (Sohl-Dickstein et al., 2015; Ho et al., 2020) is a generative model which defines a forward process to slowly add random noises to clean data $\boldsymbol{x}_0 \sim p(\boldsymbol{x})$ and a reverse process to generate desired results from random noises $\boldsymbol{\epsilon} \sim \mathcal{N}(\boldsymbol{0}, \boldsymbol{I})$ within $T$ time-steps:

$$q(\boldsymbol{x}_t|\boldsymbol{x}_{t-1}) = \mathcal{N}(\boldsymbol{x}_t; \sqrt{\alpha_t}\boldsymbol{x}_{t-1}, (1 - \alpha_t)\boldsymbol{I}), \tag{11}$$

$$p_{\boldsymbol{\theta}}(\boldsymbol{x}_{t-1}|\boldsymbol{x}_t) = \mathcal{N}(\boldsymbol{x}_{t-1}; \boldsymbol{\mu}_{\boldsymbol{\theta}}(\boldsymbol{x}_t, t), \sigma_t^2 \boldsymbol{I}), \tag{12}$$

where $\alpha_t$ and $\sigma_t$ are calculated by a pre-defined scale factor $\beta_t$, and $\boldsymbol{\mu}_{\boldsymbol{\theta}}(\boldsymbol{x}_t, t)$ is calculated by $\boldsymbol{x}_t$ and the noise $\boldsymbol{\epsilon}_{\boldsymbol{\theta}}(\boldsymbol{x}_t, t)$ predicted by a neural network, which is optimized with prediction loss:

$$\mathcal{L} = \mathbb{E}_{\boldsymbol{x}_t, \boldsymbol{\epsilon}, t} \left[ w(t) || \boldsymbol{\epsilon}_{\boldsymbol{\theta}}(\boldsymbol{x}_t, t) - \boldsymbol{\epsilon} ||_2^2 \right], \tag{13}$$

where $w(t)$ is a weighting function that depends on the time-step $t$. Recently, text-to-image diffusion models achieve impressive success in text-guided image generation by learning $\boldsymbol{\epsilon}_{\boldsymbol{\theta}}(\boldsymbol{x}_t, \boldsymbol{y}, t)$ conditioned by the text prompt $\boldsymbol{y}$. Furthermore, classifier-free guidance (CFG) (Ho & Salimans, 2022) is widely leveraged to improve the quality of results via a guidance scale parameter $\omega$:

$$\hat{\boldsymbol{\epsilon}}_{\boldsymbol{\theta}}(\boldsymbol{x}_t, \boldsymbol{y}, t) = (1 + \omega)\boldsymbol{\epsilon}_{\boldsymbol{\theta}}(\boldsymbol{x}_t, \boldsymbol{y}, t) - \omega\boldsymbol{\epsilon}_{\boldsymbol{\theta}}(\boldsymbol{x}_t, t), \tag{14}$$

**Score Distillation Sampling** (SDS) is proposed by (Poole et al., 2022) to create 3D contents from given text prompts by distilling 2D images prior from a pre-trained diffusion model to a differentiable 3D representation. Concretely, the image $\boldsymbol{x} = g(\boldsymbol{\phi})$ is rendered by a differentiable generator $g$ and a representation parameterized by $\boldsymbol{\phi}$, and the gradient is calculated as:

$$\nabla_{\boldsymbol{\phi}}\mathcal{L}_{\text{SDS}}(\boldsymbol{\theta}, \boldsymbol{x}) = \mathbb{E}_{t, \boldsymbol{\epsilon}} \left[ w(t)(\hat{\boldsymbol{\epsilon}}_{\boldsymbol{\theta}}(\boldsymbol{x}_t, \boldsymbol{y}, t) - \boldsymbol{\epsilon}) \frac{\partial \boldsymbol{x}}{\partial \boldsymbol{\phi}} \right]. \tag{15}$$

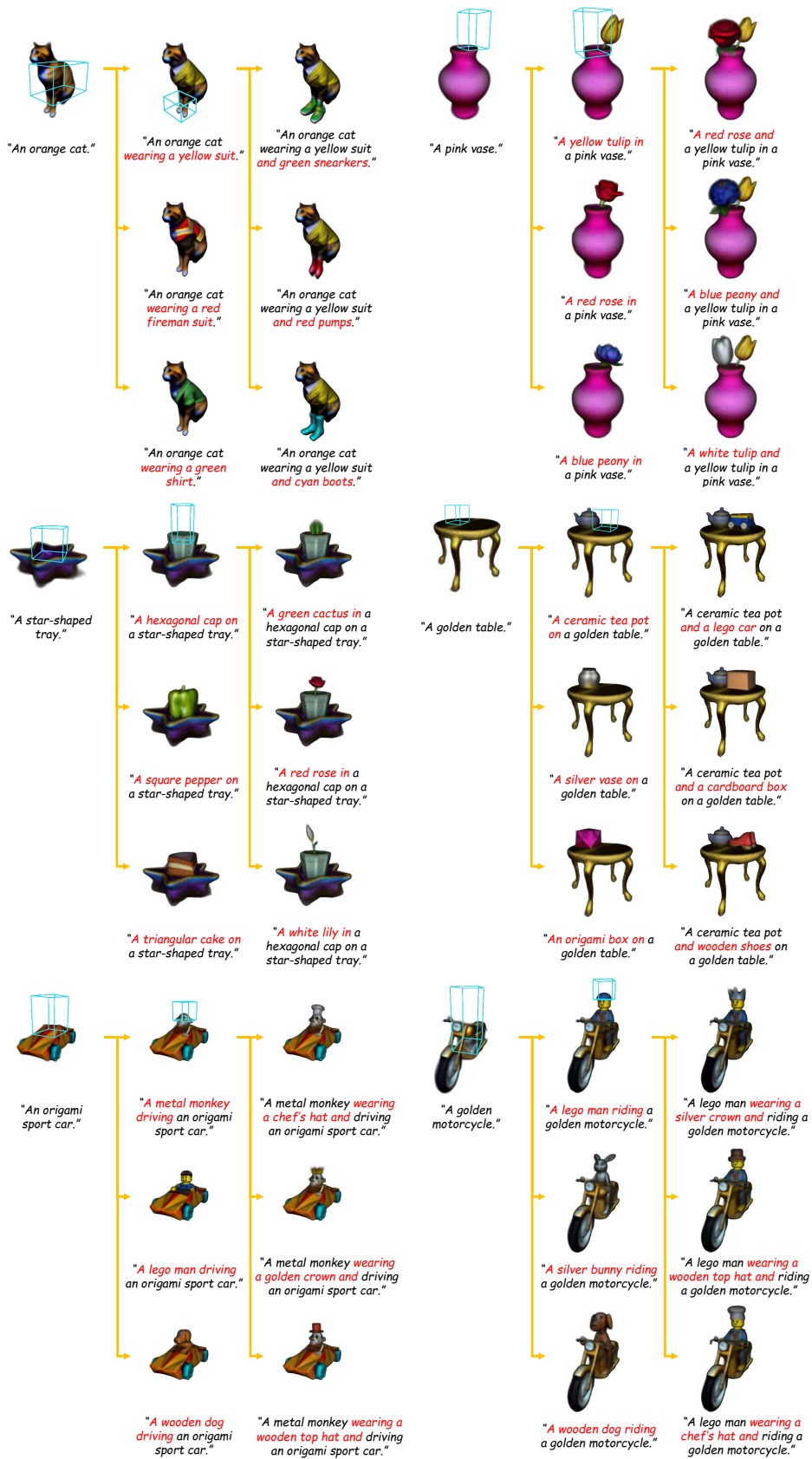

Figure 16: More progressive editing results created with our Progressive3D based on DreamTime.

Table 4: Detailed prompt list of CSP-100. (Part 1)

| Index | Prompt |
|---|---|
| **Color** (*i.e.*, two interacted objects binding with different colors) | |
| 1 | an orange cat wearing a yellow suit |
| 2 | an orange cat wearing a green shirt |
| 3 | an orange cat wearing a red fireman uniform |
| 4 | a red rose in a pink vase |
| 5 | a blue peony in a pink vase |
| 6 | a yellow tulip in a pink vase |
| 7 | a pair of red sneakers on a blue chair |
| 8 | a stack of green books on a blue chair |
| 9 | a purple gift box on a blue chair |
| 10 | a red cake in a yellow tray |
| 11 | a blue spoon in a yellow tray |
| 12 | a pair of green chopsticks in a yellow tray |
| 13 | a green vase on a red desk |
| 14 | a pair of blue sneakers on a red desk |
| 15 | a yellow tray on a red desk |
| **Shape** (*i.e.*, two interacted objects binding with different shapes) | |
| 16 | a round gift box on a hexagonal table |
| 17 | a triangular cake on a hexagonal table |
| 18 | a square tray on a hexagonal table |
| 19 | a hexagonal cup on a round cabinet |
| 20 | a triangular sandwich on a round cabinet |
| 21 | a square bowl on a round cabinet |
| 22 | a hexagonal cup on a star-shaped tray |
| 23 | a triangular cake on a star-shaped tray |
| 24 | a square pepper on a star-shaped tray |
| 25 | a model of a round house with a hexagonal roof |
| 26 | a model of a round house with a square roof |
| 27 | a model of a round house with a spherical roof |
| **Material** (*i.e.*, two interacted objects binding with different materials) | |
| 28 | a lego man riding a golden motorcycle |
| 29 | a silver bunny riding a golden motorcycle |
| 30 | a wooden dog riding a golden motorcycle |
| 31 | a lego man driving an origami sport car |
| 32 | a metal monkey driving an origami sport car |
| 33 | a wooden dog driving an origami sport car |
| 34 | a ceramic tea pot on a golden table |
| 35 | a silver vase on a golden table |
| 36 | an origami box on a golden table |
| 37 | a model of a silver house with a golden roof |
| 38 | a model of a silver house with a wooden roof |
| 39 | a model of a silver house with a bronze roof |
| 40 | a lego tank with a golden gun |
| 41 | a lego tank with a silver gun |
| 42 | a lego tank with a wooden gun |
| **Composition** (*i.e.*, more than two interacted objects binding with different attributes) | |
| 43 | an orange cat wearing a yellow suit and green sneakers |
| 44 | an orange cat wearing a yellow suit and red pumps |
| 45 | an orange cat wearing a yellow suit and cyan boots |
| 46 | an orange cat wearing a yellow suit and green sneakers and cyan top hat |
| 47 | an orange cat wearing a yellow suit and green sneakers and pink cap |
| 48 | an orange cat wearing a yellow suit and green sneakers and red chef's hat |
| 49 | a blue peony and a yellow tulip in a pink vase |
| 50 | a red rose and a yellow tulip in a pink vase |

Table 5: Detailed prompt list of CSP-100. (Part 2)

| Index | Prompt |
|---|---|
| 51 | a white tulip and a yellow tulip in a pink vase |
| 52 | a purple rose and a red rose and a yellow tulip in a pink vase |
| 53 | a blue peony and a red rose and a yellow tulip in a pink vase |
| 54 | a white tulip and a red rose and a yellow tulip in a pink vase |
| 55 | a golden cat on a stack of green books on a blue chair |
| 56 | a wooden bird on a stack of green books on a blue chair |
| 57 | a lego car on a stack of green books on a blue chair |
| 58 | a blue candle on a red cake in a yellow tray |
| 59 | a green spoon on a red cake in a yellow tray |
| 60 | a pair of cyan chopsticks on a red cake in a yellow tray |
| 61 | a cyan cake on a yellow tray on a red desk |
| 62 | a ceramic tea pot on a yellow tray on a red desk |
| 63 | a green apple on a yellow tray on a red desk |
| 64 | a square cake on a square tray on a hexagonal table |
| 65 | a round cake on a square tray on a hexagonal table |
| 66 | a triangular sandwich on a square tray on a hexagonal table |
| 67 | a green apple in a hexagonal cup on a round cabinet |
| 68 | a pink peach in a hexagonal cup on a round cabinet |
| 69 | a yellow pineapple in a hexagonal cup on a round cabinet |
| 70 | a red rose in a hexagonal cup on a star-shaped tray |
| 71 | a white lily in a hexagonal cup on a star-shaped tray |
| 72 | a green cactus in a hexagonal cup on a star-shaped tray |
| 73 | a lego man wearing a chef's hat and riding a golden motorcycle |
| 74 | a lego man wearing a wooden top hat and riding a golden motorcycle |
| 75 | a lego man wearing a silver crown and riding a golden motorcycle |
| 76 | a metal monkey wearing a golden crown and driving an origami sport car |
| 77 | a metal monkey wearing a chef's hat and driving an origami sport car |
| 78 | a metal monkey wearing a wooden top hat and driving an origami sport car |
| 79 | a ceramic tea pot and a lego car on a golden table |
| 80 | a ceramic tea pot and an origami box on a golden table |
| 81 | a ceramic tea pot and a cardboard box on a golden table |
| 82 | a ceramic tea pot and a pair of wooden shoes on a golden table |
| 83 | a ceramic tea pot and an origami box and a green apple on a golden table |
| 84 | a ceramic tea pot and an origami box and a yellow tray on a golden table |
| 85 | a ceramic tea pot and an origami box and a stack of blue books on a golden table |
| 86 | a model of a silver house with a golden roof beside an origami coconut tree |
| 87 | a model of a silver house with a golden roof beside a wooden car |
| 88 | a model of a silver house with a golden roof beside a lego man |
| 89 | a lego tank with a golden gun and a red flying flag |
| 90 | a lego tank with a golden gun and a blue flying flag |
| 91 | a lego tank with a golden gun and a yellow flying flag |
| 92 | a model of a round house with a spherical roof on a hexagonal park |
| 93 | a model of a round house with a spherical roof on a square park |
| 94 | a model of a round house with a spherical roof on an oval park |
| 95 | an astronaut holding a red rifle and riding a green origami motorcycle |
| 96 | an astronaut holding a red rifle and riding a cyan scooter |
| 97 | an astronaut holding a red rifle and riding a golden motorcycle |
| 98 | an astronaut holding a red rifle and riding a green origami motorcycle and wearing a blue top hat |
| 99 | an astronaut holding a red rifle and riding a green origami motorcycle and wearing a cyan chef's hat |
| 100 | an astronaut holding a red rifle and riding a green origami motorcycle and wearing a pink cowboy hat |

