# OpenReview forum: "Progressive3D: Progressively Local Editing for Text-to-3D Content Creation with Complex Semantic Prompts"
_ICLR.cc/2024/Conference — ICLR 2024 poster_

### Official Review · Reviewer_EZDr · 2023-10-31

**Soundness:** 3 good
**Presentation:** 3 good
**Contribution:** 2 fair
**Rating:** 6
**Confidence:** 3

**Summary:**

This paper presents a locally progressive method for text-driven generation of semantically complex prompts. Existing methods cannot faithfully generate the complex prompts directly. Thus, this work takes an iterative approach: the complex prompt is broken up into segments and each segment is iteratively added to the 3D representation. However, directly optimizing for the new segment of the prompt each iteration leads to a muddled result where attributes bleed together. Thus, this work also conditions on a user specified edit region. This allows the method to only optimize a local region, thus preserving the rest of the 3D representation.

**Strengths:**

- Generation with complex prompts is very challenging and existing methods struggle on this task, while this work excels.
- The semantic delta loss is an interesting contribution that is important for editing.
- Explicitly defines a region in which the edits can take place to ensure preservation of the existing model.
- The paper compares to numerous existing approaches for text-to-image generation showing superior performance on complex prompts and gives a thorough ablation of the components of the method.
- Clear presentation: the paper is well written and makes good use of experiments/figures to support its claims. Figure 2 is especially helpful for understanding this approach.

**Weaknesses:**

- The local region for each edit must be manually entered by the user. This slightly limits the intuitive, easy-to-use nature of this work as compared to most other purely text-driven approaches.
- If I am understanding correctly (see question for more details), the edit region can only be defined as an axis-aligned bounding box. This seems like it could be problematic for edits that do not fit nicely into an axis-aligned box.
- Limited comparisons to relevant existing work. There exist methods for focusing on different parts of the text prompt (Attend and Excite [1]) for 2D image generation and editing. These methods have been shown to address issues with attribute binding and “catastrophic neglect.” It would be helpful to see how a baseline performs using these approaches as the 2D model used for distillation. Additionally, DreamEditor [2] enables local editing on NeRFs using an explicit edit region. The region is inferred from the text description using attention maps from the diffusion model. A good baseline would be to use DreamEditor iteratively on each progressive edit as this still gives an explicit edit region, but does not require a user input bounding box.

References:
[1] Chefer, Hila, Yuval Alaluf, Yael Vinker, Lior Wolf, and Daniel Cohen-Or. "Attend-and-excite: Attention-based semantic guidance for text-to-image diffusion models." ACM Transactions on Graphics (TOG) 42, no. 4 (2023): 1-10.
[2] Zhuang, Jingyu, Chen Wang, Lingjie Liu, Liang Lin, and Guanbin Li. "DreamEditor: Text-Driven 3D Scene Editing with Neural Fields." arXiv preprint arXiv:2306.13455 (2023).

**Questions:**

- Since the user can only input the box center and the lengths of the box along each axis, it seems that the box will always be axis aligned. Is this not problematic for certain edits that do not line up well with the axes?
- It would be helpful to clarify more how the semantic delta loss differs from (Armandpour et al.) [4]

References:
[4] Armandpour, Mohammadreza, Huangjie Zheng, Ali Sadeghian, Amir Sadeghian, and Mingyuan Zhou. "Re-imagine the Negative Prompt Algorithm: Transform 2D Diffusion into 3D, alleviate Janus problem and Beyond." arXiv preprint arXiv:2304.04968 (2023).

**Details Of Ethics Concerns:**

No ethical concerns.

---

> ### Author Response · Authors · 2023-11-22
> **Response to Reviewer EZDr**
>
> We appreciate your time devoted to reviewing this paper and your constructive suggestions! Here are our detailed replies to your questions. We humbly expect you can check the replies and reconsider the decision.
>
> > **Q1:** The boulding box is inflexable for region selection compared to text-driven region selection.
>
> **A1:**
> We underline that our Progressive3D framework supports **different region definitions** including 3D forms (3D boxes, custom meshes) and 2D forms (2D segmentation and attention maps) since the corresponding 2D region mask of each view can be obtained.
> (See in **Sec.3.2** and **Appx.A.3**). We adopt 3D boxes as region definitions in **Sec.3** for brevity.
> Differing from existing 3D editing methods, our $\mathcal{L}_{consist}$ is imposed on 2D rendered images and is effective for various 3D representations and region definitions (See discussion in **Appx.A.5**).
> We have provided editing samples in **Fig.11** in the revised paper that demonstrates Progressive3D is available with region defined by multiple 3D boxes, custom meshes, and 2D segmentation queried by the provided keyword (**text-driven region selection**).
>
> Noticing that user-provided masks and automatically calculated regions of interest are both widely used in 2D repainting/editing methods, we believe that defining a specific region instead of leveraging the inaccurate attention maps is necessary in many scenarios, and our framework also supports attention maps as the region definition.
>
> > **Q2:** The edit region can only be defined as an axis-aligned bounding box.
>
> **A2:**
> As described in **Sec.3.3**, the 3D bounding box can be rotated arbitrarily with the corresponding rotate matrix $R$, and the 3D bounding box and its rotation matrix can be obtained conveniently through a 3D GUI.
>
> > **Q3:** More comparison with other methods.
>
> **A3:**
> We have provided the qualitative and quantitative comparison with DreamTime-based baselines including DreamTime+CEBM and DreamTime+A&E on CSP-100 in **Fig.6** and **Tab.1** in the revised paper.
> Since composing T2I methods including CEBM [1] and A&E [2] are not designed for composing 3D content generation, combining composing T2I methods with DreamTime brings limited performance improvements (even regression), while DreamTime+Progressive3D is superior to all DreamTime-based baselines.
> We analyze that although compositional T2I models produce more desirable results with complex prompts than normal T2I methods in each view, the inconsistency among views is still significant, leading to low-quality results.
> However, Progressive3D reduces the generation difficulty since the current 3D content has been achieved ideally with the user-provided region definitions and proposed OSCS technique in each editing step.
>
> Compared to DreamEditor, our Progressive3D theoretically performs better in the following aspects: **(1) Generalization:** Their method is carefully designed on mesh-based neural fields and costs extra time and quality loss for distilling other 3D representations to mesh-based neural fields. However, our Progressive3D is general to various 3D representations and region definitions if the corresponding projected depth and opacity can be obtained by rasterization or volume rendering.
> **(2) Geometry variation:** Since their editable regions are defined on existing mesh, DreamEditor may suffer challenges when creating additional objects interacting with existing objects with rapid geometry variation, *e.g.*, editing *"an astronaut"* to *"an astronaut riding a motorcycle"*.
> However, our Progressive3D proposes an initialization constraint $\mathcal{L}_{initial}$ and effectively tackles the desired rapid geometry variation (See in **Fig.3(d)&(f)** in revised paper).
> **(3) Semantic Composition:** There are no specially designed modules to tackle the complicated semantics in DreamEditor, and multiple objects with different attributes may cause issues in text-driven region location and optimization directions.
> The comparison with DreamEditor is not implemented due to the time limits, and we will add more comparison results and discussions in the final version.

---

> ### Author Response · Authors · 2023-11-22
> **Response to Reviewer EZDr: Part 2**
>
> > **Q4:** The difference between our OSCS and [3].
>
> **A4:**
> The commonality of OSCS and [3] is we both leverage the perpendicular component between different semantic components as the overlapped semantics. However, the design goal of OSCS and [3] is different, leading to differences in expression forms.
> Concretely, we re-formulate their $\hat{\epsilon}^{Prep}\_\theta$ as $\hat{\epsilon}^{Prep}\_\theta(x_t, y_s, y_t, t)=\epsilon\_\theta(x_t, t) + \omega(\Delta\epsilon^s\_\theta+w\Delta\epsilon^{prep}\_\theta)$, and our $\hat{\epsilon}^{OSCS}\_\theta$ as $\hat{\epsilon}^{OSCS}\_\theta(x_t, y_s, y_t, t)=\epsilon_\theta(x_t, t) + \omega(\frac{1}{W}\Delta\epsilon^{proj}\_\theta+\Delta\epsilon^{prep}\_\theta).$
> Therefore, their component of $y_s$ is the main optimization orientation and the perpendicular component is leveraged to supply auxiliary information, such as directions (*e.g.*, front, side, back).
> However, we mainly focus on the perpendicular component for paying more attention to generating newly mentioned objects or attributes, and the projection component is weighted for harmonious interactives between target objects and generated objects.
>
> ----
>
> **Reference:**
>
> [1] Liu, Nan, et al. "Compositional visual generation with composable diffusion models." European Conference on Computer Vision. Cham: Springer Nature Switzerland, 2022.
>
> [2] Chefer, Hila, et al. "Attend-and-excite: Attention-based semantic guidance for text-to-image diffusion models." ACM Transactions on Graphics (TOG) 42.4 (2023): 1-10.
>
> [3] Armandpour, Mohammadreza, et al. "Re-imagine the Negative Prompt Algorithm: Transform 2D Diffusion into 3D, alleviate Janus problem and Beyond." arXiv preprint arXiv:2304.04968 (2023).

---

> > ### Comment · Reviewer_EZDr · 2023-11-23
> > **response to rebuttal**
> >
> > The authors addressed my concerns on a comparison to Attend and Excite and on the flexibility of the edit region. The argument for why DreamEditor theoretically would perform worse is convincing, but I would want to see an actual comparison to be fully convinced. Specifically,  “geometry variation” could still be achieved with DreamEditor even with the localization contained to the surface of the mesh and complicated semantics (“semantic composition”) could be tackled with DreamEditor using the same progressive approach. While there is limited time for the rebuttal so not all experiments can be run, I would encourage the authors to include DreamEditor comparisons that touch on these points in a final draft. Additionally, while the use of text-driven segmentation to provide the edit region automates the edit region selection, the overall process is still not fully automatic as the decomposition still requires human intervention. After reviewing the rebuttal and revised submission, I am increasing my score to 6.

---

> > > ### Author Response · Authors · 2023-11-23
> > > **Response to Reviewer EZDr**
> > >
> > > We really appreciate your constructive suggestions and reconsidered rating! We will add more comparison results and discussions in the final version. Thanks again!

---

### Official Review · Reviewer_GY6s · 2023-10-31

**Soundness:** 3 good
**Presentation:** 3 good
**Contribution:** 3 good
**Rating:** 6
**Confidence:** 5

**Summary:**

The paper introduces a method for progressively generating intricate 3D content. Each generation phase progressively generates local content using progressive semantic prompts. To maintain the consistency of the progressive generation, a consist loss is employed. Additionally, an initialization loss is utilized to swiftly generate content in selected areas. Furthermore, the proposed OVERLAPPED SEMANTIC COMPONENT SUPPRESSION ensures that each progressive generation optimizes towards additional semantic prompts.

**Strengths:**

1.The progressive approach is indeed a straightforward and effective method for generating complex 3D content. This ensures that each local element receives accurate optimization guidance.
2."OVERLAPPED SEMANTIC COMPONENT SUPPRESSION" can effectively optimize progressive 3D content in alignment with additional semantic prompts.
3.The concept of "Initial Loss" contributes to achieving a stable and high-quality 3D content generation within 3D bounding box.

**Weaknesses:**

1. Further comparisons with other methods for achieving complex semantic prompt-driven 3D content generation are lacking.
2. The overall pipeline is relatively straightforward and simplified. Independently generating each local content and optimizing it after the combining each local 3D content with 3D bounding boxes may yield improved results, particularly for the prompt like some object is on a tabletop.

**Questions:**

1.Is the progressive3D approach the only way to generate intricate 3D content? I want to see more comparative experiments to achieve a fairer comparison. For instance, one might first employ a complex semantic prompt to produce consistent images, then use these images to generate the corresponding 3D content. Alternatively, one could generate each object individually and subsequently merge them based on their bounding boxes, then the aggregated 3D content could be optimized to achieve a harmonious result. I am interested in understanding how to demonstrate that the progressive approach is a crucial and effective method for generating complex 3D content.

2. How to resolve conflicts between subsequent generated results and earlier ones, such as an astronaut sitting on a red chair, when the first step generates a standing astronaut? Can the model optimize the transition from a standing astronaut to a sitting one?

3.In the context of complex semantic prompts, does the order of prompt input have an impact on the generated results, and is it necessary to engage in simple semantic prompt planning?

---

> ### Author Response · Authors · 2023-11-22
> **Response to Reviewer GY6s**
>
> We appreciate your time devoted to reviewing this paper and your constructive suggestions! Here are our detailed replies to your questions. We humbly expect you can check the replies and reconsider the decision.
>
> > **Q1:** More comparison with other methods.
>
> **A1:**
> We have provided the qualitative and quantitative comparison with DreamTime-based baselines including DreamTime+CEBM and DreamTime+A&E on CSP-100 in **Fig.6** and **Tab.1** in the revised paper.
> Since composing T2I methods including CEBM [1] and A&E [2] are not designed for composing 3D content generation, combining composing T2I methods with DreamTime brings limited performance improvements (even regression), while DreamTime+Progressive3D is superior to all DreamTime-based baselines.
> We analyze that although compositional T2I models produce more desirable results with complex prompts than normal T2I methods in each view, the inconsistency among views is still significant, leading to low-quality results.
> However, Progressive3D reduces the generation difficulty since the current 3D content has been achieved ideally with the user-provided region definitions and proposed OSCS technique in each editing step.
>
> > **Q2:** The overall pipeline is relatively straightforward and simplified.
>
> **A2:**
> We consider that our Progressive3D is simplified but effective.
> Concretely, our Content Consistency Constraint is designed for the generalization on various text-to-3D methods and region definitions, our Content Initialization Constraint is proposed to reduce the generation difficulty when the geometry is changed rapidly, and our Overlapped Semantic Component Suppression is the core for progressive additional editing especially when the source prompt is already complicated.
>
> We recognize that there is an alternative solution *i.e.,* separately generates objects and places them with some region definitions when the mentioned objects simply interact. However, Progressive3D can generate objects with complicated interacts, as shown in **Fig.7&10&11** in revised paper, which cannot be achieved by placing individual objects according to 3D boxes directly.
>
> > **Q3:** Other possible ways to generate intricate 3D content.
>
> **A3:**
> We give the discussion of our Progressive3D and the paths you mentioned (**1**.Generating consistent multi-view images with complex prompts and **2**.generating coarse objects first and fine-tuning for harmony results) here.
>
> For path **1**, the requirement of generating images with complex prompts conflicts with the multi-view consistency since the images with complex content tend to be inconsistent, and the prediction of geometry information including depths and normals is more difficult when the image is complicated.
> Furthermore, we highlight that our Progressive3D is **orthogonal** to the improvement of multi-view images generator since MVDream (Submitted to ICLR2024) increases its 3D content generation capacity by leveraging a powerful multi-view images generator and we verified Progressive3D is available with MVDream and achieves impressive results in revised paper.
>
> In addition, we consider that Progressive3D and path **2** can be separately seen as the 3D lifting of 2D progressive editing and 2D layout generation, where progressive editing is more controllable and layout generation is more time efficient.
> As discussed in **Appx.A.4**, we have noticed such potential direction (3D layout generation) and will further explore it in future works.

---

> ### Author Response · Authors · 2023-11-22
> **Response to Reviewer GY6s: Part 2**
>
> > **Q4:** The conflicts between source 3D content and the desired editing results.
>
> **A4:**
> Generating harmonious 3D content with geometry variation (resolving conflicts between prompts) is a significant superiority compared to existing 3D editing methods including DreamEditor and FocalDreamer.
> We achieve such a goal by introducing $\mathcal{L}_{initial}$ and OSCS and the editing results with geometry variation (turn the standing pose into riding or sitting) are shown in **Fig.7&10&11** in revised paper.
>
> > **Q5:** The impact of the prompt order for generated results.
>
> **A5:**
> We have provided discussion and visual results in **Fig.10** in revised paper.
> We find that different object generating orders in Progressive3D typically result in correct 3D content consistent with the complex prompts.
> However, the content details of the final content are impacted by created objects since Progressive3D is a local editing chain started from the source content.
> With different generating orders, Progressive3D creates 3D content with different details while they are both consistent with the prompt *''*An astronaut sitting on a wooden chair''*, as shown in **Fig.10**.
>
> ----
>
> **Reference:**
>
> [1] Liu, Nan, et al. "Compositional visual generation with composable diffusion models." European Conference on Computer Vision. Cham: Springer Nature Switzerland, 2022.
>
> [2] Chefer, Hila, et al. "Attend-and-excite: Attention-based semantic guidance for text-to-image diffusion models." ACM Transactions on Graphics (TOG) 42.4 (2023): 1-10.

---

> > ### Comment · Reviewer_GY6s · 2023-11-23
> > **Detailed responses**
> >
> > Thank you for your comprehensive experiments and detailed responses, which excellently addressed my questions.

---

> > > ### Author Response · Authors · 2023-11-23
> > > **Would you mind reconsidering your rating?**
> > >
> > > Thanks for your reply! It would be very appreciated if you reconsider your rating since we have excellently addressed your concerns. Thanks again!

---

### Official Review · Reviewer_dBWG · 2023-11-01

**Soundness:** 3 good
**Presentation:** 3 good
**Contribution:** 3 good
**Rating:** 6
**Confidence:** 4

**Summary:**

This paper tackles the challenge of aligning complex prompts with generated 3D assets by employing a progressive framework that decomposes the generation process into multiple local editing tasks. By doing so, the authors achieve the generation of semantically precise 3D assets. Additionally, the paper introduces a novel dataset designed to evaluate the outcomes of compositional Text-to-3D generation.

**Strengths:**

1. This paper is well-written, with a clear structure and easily understandable language.

2. The results demonstrate effective composition of objects with different attributes and relationships.

3. The proposed dataset and evaluation metrics explore Text-to-3D benchmark in terms of composition and relationships, providing valuable insights for the community.

**Weaknesses:**

1. The main concern is the heavy reliance on human involvement throughout the pipeline. Users are required to provide prompt divisions and bounding boxes, and the process seems user-unfriendly, as users have to wait for the previous generation to finish before providing the next bounding box prompt.

2. The paper mentions that current T2I diffusion models often struggle with complex prompts. However, there are existing methods [1,2,3] that address this problem. It would be beneficial to discuss why the authors did not directly utilize these methods, as it seems more straightforward and would save human labor. This discussion is currently missing from the paper.

3. Figures 7 and 10 show inconsistencies with the claim that undesired regions remain unchanged. The leg of the astronaut turns green, and the foot is missing after adding the prompt "and riding a red motorcycle."

4. It would be more convincing if the paper showcased additional results (quantitative and qualitative) based on Fantasia3D. Given the low resolution of image space supervision in DreamTime and DreamFusion, the generated 3D assets appear blurry. Demonstrating the significant improvements offered by this method in more sophisticated Text-to-3D approaches would reinforce the paper's claims.

5. Including an ablation study on the last term in the consistency loss (the one that imposes the empty region to be blank) would strengthen the paper's arguments.

[1] Compositional visual generation with composable diffusion models, ECCV, 2022

[2] Training-Free Structured Diffusion Guidance for Compositional Text-toImage Synthesis, ICLR, 2023

[3] Attend-and-excite: Attention-based semantic guidance for text-to-image diffusion models, ICLR, 2023

**Questions:**

Please refer to the weakness section.

---

> ### Author Response · Authors · 2023-11-22
> **Response to Reviewer dBWG**
>
> We appreciate your time devoted to reviewing this paper and your constructive suggestions! Here are our detailed replies to your questions. We humbly expect you can check the replies and reconsider the decision.
>
> > **Q1:** Seems user-unfriendly including providing 3D regions, composing prompts and creating 3D content progressively.
>
> **A1:**
> Although we present many long-chain progressive editing processes to demonstrate our Progressive3D supports multi-step complex editing, users might edit the source 3D content only once or twice to achieve desired 3D content in practice.
> As discussed in **Appx.A.1**, we consider Progressive3D accords with user realistic usage pipeline, *i.e.,* creating a primary object first, then adjusting its attribute or adding more related objects, and Progressive3D in practice is **flexible** for different user goals.
> For instance in **Fig.9** in revised paper, we desire to create the 3D content consistent with the prompt *''An astronaut wearing a green top hat and riding a red horse''*.
> We find that MVDream (Submitted to ICLR2024) fails to create the precise result while generating *''An astronaut riding a red horse''* correctly.
> Thus the desired content can be achieved by editing *''An astronaut riding a red horse''* within one-step editing, instead of starting from *''an astronaut''*.
>
> Furthermore, we underline that our Progressive3D framework supports **different region definitions** including 3D forms such as 3D boxes, custom meshes and 2D forms such as 2D segmentation and attention maps since the corresponding 2D region mask of each view can be obtained.
> (See in **Sec.3.2** and **Appx.A.3**).
> We have provided editing samples in **Fig.11** in the revised paper that demonstrate Progressive3D is available with region defined by multiple 3D boxes, custom meshes, and 2D segmentation queried by the provided keyword, which can be convenient for region selection by users.
>
> > **Q2:** Discussion about directly using T2I models designed for complex prompts.
>
> **A2:**
> We have provided the qualitative and quantitative comparison with DreamTime-based baselines including DreamTime+CEBM and DreamTime+A&E on CSP-100 in **Fig.6** and **Tab.1** in the revised paper.
> Since composing T2I methods including CEBM [1] and A&E [2] are not designed for composing 3D content generation, combining composing T2I methods with DreamTime brings limited performance improvements (even regression), while DreamTime+Progressive3D is superior to all DreamTime-based baselines.
> We analyze that although compositional T2I models produce more desirable results with complex prompts than normal T2I methods in each view, the inconsistency among views is still significant, leading to low-quality results.
> However, Progressive3D reduces the generation difficulty since the current 3D content has been achieved ideally with the user-provided region definitions and proposed OSCS technique in each editing step.
>
> > **Q3:** More results for higher-quality 3D assets.
>
> **A3:**
> We highlight that our Progressive3D is a **general framework** for progressive local editing for various text-to-3D methods driven by different 3D neural representations.
> Thus the quality of our generated 3D content can be improved with the development of text-to-3D area.
>
> **Fig.1&5** in revised paper demonstrate that our editing can produce high-quality 3D content based on Fantasia3D, and our method is also proved effective on other high-quality text-to-3D method like ProlificDreamer[3].
> However, 3D results created by high-quality methods, especially ProlificDreamer severely suffered from **geometric inaccuracy problems** including multi-faces and floating noisys, even though their textures are impressive in a single view.
> To generate stable results for quantitative comparison, we have to choose DreamTime as the baseline since DreamTime is relatively robust for different prompts and seeds.
>
> We are glad to notice that concurrent works named MVDream (Submitted to ICLR2024) achieve impressive 3D generation performance, and we have verified that Progressive3D works well on MVDream, which further demonstrates the generalization of our Progressive3D. Please check our revised paper for more visual results.

---

> ### Author Response · Authors · 2023-11-22
> **Response to Reviewer dBWG: Part 2**
>
> > **Q4:** Figure 7&10 show inconsistence.
>
> **A4:**
> We claim that the changes of astronaut legs & feet are allowable since we omit the 3D bounding box in **Fig.7&10** in the original paper for brevity.
> To achieve reasonable geometry variation with *'riding a motorcycle''*, *i.e.,* changing the astronaut from standing to riding, we have to select the legs & feet in the 3D box for editing.
> The quality reduction is mainly caused by the poor generation capacity of TextMesh.
> We further provided similar samples (turn the standing pose into riding or sitting) in **Fig.7&10&11** in revised paper and demonstrate that Progressive3D can produce harmonious geometry and appearance variations by combining stronger text-to-3D generation methods such as MVDream.
>
>
> > **Q5:** Ablation study on the last term in the consistency loss.
>
> **A5:**
> We have provided a discussion and visual ablation in **Appx.C.1** and **Fig.15** in revised paper.
> We highlight that we divide the $\mathcal{L}_{consist}$ into a content term and an empty term to avoid mistakenly treating backgrounds as a part of foreground objects.
> The visual results demonstrate that mistakenly treating backgrounds as a part of foreground objects leads to significant floating.
>
> ----
>
> **Reference:**
>
> [1] Liu, Nan, et al. "Compositional visual generation with composable diffusion models." European Conference on Computer Vision. Cham: Springer Nature Switzerland, 2022.
>
> [2] Chefer, Hila, et al. "Attend-and-excite: Attention-based semantic guidance for text-to-image diffusion models." ACM Transactions on Graphics (TOG) 42.4 (2023): 1-10.
>
> [3] Wang, Zhengyi, et al. "ProlificDreamer: High-Fidelity and Diverse Text-to-3D Generation with Variational Score Distillation." arXiv preprint arXiv:2305.16213 (2023).

---

> > ### Comment · Reviewer_dBWG · 2023-12-04
> > **response to rebuttal**
> >
> > I have carefully reviewed the new experiments and analysis. The authors have addressed most of my concerns. I believe composition is an important problem for the next stage of text-to-3D generation, and the authors have provided a promising way to do this.

---

### Official Review · Reviewer_S6PX · 2023-11-01

**Soundness:** 2 fair
**Presentation:** 3 good
**Contribution:** 3 good
**Rating:** 5
**Confidence:** 4

**Summary:**

This paper proposes a general framework named Progressive3D for correctly generating 3D content when the given prompt is complex in semantics. Progressive3D decomposes the difficult creation process into a series of local editing steps and progressively generates the aiming object with binding attributes. Experiments conducted on complex prompts in CSP-100 demonstrate that the proposed Progressive3D can create 3D content consistent with complex prompts. The motivation of generate correct 3D content for a complex prompt in semantics is good, but the solution is inflexible. Because users need to provide 3D bounding box prompt for each prompt, which is inflexible and difficult to define.

**Strengths:**

- Progressive3D can create precise 3D content prompted with complex semantics by decomposing a difficult generation process into a series of local editing steps.
- Progressive3D could be incorporated into various text-to-3D methods driven by different 3D neural representations.

**Weaknesses:**

- The quality of generated 3D objects is poor.
- The proposed method requires the 3D bounding box as the input, which is inflexible.
- It is difficult for Progressive3D to change the attribute of the generated 3D objects, such as changing red to blue or metal to wood. If we want to edit the attribute, we might need to train the model case by case.
- Only one dataset was used in the experiments.

**Questions:**

- How long does it take for complex prompts?
- Given a complex prompt, how to decompose the complex text, automatically or manually? And How many steps are required?
- How can we provide the 3D bounding box prompt, I think it is difficult. In my opinion, I think the 3D bounding box limits the application of the proposed method.
- Can you report the CLIP-Score?

---

> ### Author Response · Authors · 2023-11-22
> **Response to Reviewer S6PX**
>
> We appreciate your time devoted to reviewing this paper and your constructive suggestions! Here are our detailed replies to your questions. We humbly expect you can check the replies and reconsider the decision.
>
> > **Q1:** The quality of generated 3D is poor.
>
> **A1:**
> We highlight that our Progressive3D is a **general framework** for progressive local editing for various text-to-3D methods driven by different 3D neural  representations.
> Thus the quality of our generated 3D content can be improved with the development of text-to-3D area.
>
> **Fig.1&5** in revised paper demonstrate that our editing can produce high-quality 3D content based on Fantasia3D, and our method is also proved effective on other high-quality text-to-3D method like ProlificDreamer[1].
> However, 3D results created by high-quality methods especially ProlificDreamer are severe suffered from **geometric inaccuracy problems** including multi-faces and floating noisys, even though their textures are impressive in a single view.
> To generate stably results for quantitative comparison, we have to choose DreamTime as the baseline since DreamTime is relatively robust for different prompts and seeds.
>
> We glad to notice that concurrent works named MVDream (Submitted to ICLR2024) achieves impressive 3D generation performance, and we have verfied that Progressive3D works well on MVDream, which further demonstrates the generalization of our Progressive3D. Please check our revised paper for more visual results.
>
>
> > **Q2:** Providing the 3D bounding box prompt is difficult and inflexible.
>
> **A2:**
> We underline that our Progressive3D framework supports **different region definitions** including 3D forms (3D boxes, custom meshes) and 2D forms (2D segmentation and attention maps) since the corresponding 2D region mask of each view can be obtained.
> (See in **Sec.3.2** and **Appx.A.3**). We adopt 3D boxes as region definitions in **Sec.3** for brevity, and the 3D bounding box can be obtained conveniently through a 3D GUI.
> Differing from existing 3D editing methods, our $\mathcal{L}_{consist}$ is imposed on 2D rendered images and is effective for various 3D representations and region definitions (See discussion in **Appx.A.5**).
> We have provided editing samples in **Fig.11** in the revised paper that demonstrates Progressive3D is available with regions defined by multiple 3D boxes, custom meshes, and 2D segmentation queried by the provided keyword.
>
> Noticing that user-provided masks and automatically calculated regions of interest are both widely used in 2D repainting/editing methods, we believe that defining a specific region instead of leveraging the inaccurate attention maps is necessary for many scenarios, and our framework also supports attention maps as the region defination.
>
> > **Q3:** Progressive3D is difficult to change the attribute of generated 3D objects.
>
> **A3:**
> We emphasize that each step in Progressive3D is a **local editing step** and Progressive3D supports both **modifying attributes of existing objects** and **creating additional objects with attributes not mentioned in source prompts from scratch** in user-selected regions.
> We have provided attribute editing results in **Fig.12** in revised paper.
> Noticing that creating additional objects with attributes not mentioned in source prompts from scratch is more difficult than editing the attributes of existing objects.
> Therefore, attribute editing costs significantly less time than additional object generation.
>
> > **Q4:** Only one dataset is used for evaluation.
>
> **A4:**
> Considering the 3D editing task with complicated prompts has not been broadly studied, each work has to design specific settings and datasets to demonstrate the effectiveness of the proposed method.
> We underline that our proposed CSP-100 is the first dataset for complex text-to-3D content generation.
> For more convincing, we have provided more results on MVDream in the revised paper.
>
> > **Q5:** Time cost for complex prompts.
>
> **A5:**
> Although we present many long-chain progressive editing processes to demonstrate our Progressive3D supports multi-step complex editing, users might edit the source 3D content only once or twice to achieve desired 3D content in practice (See in **Appx.A.1**).
> Similar to many 2D repainting/editing methods, our one editing step costs a similar time to one generation of base methods from scratch (See in **Appx.B.3**), which means the time efficiency of Progressive3D can be further improved simultaneously with the text-to-3D methods.

---

> ### Author Response · Authors · 2023-11-22
> **Response to Reviewer S6PX: Part2**
>
> > **Q6:** How to decompose the complex prompt.
>
> **A6:**
> As discussed in **Appx.A.1**, the Progressive3D editing in practice is **flexible** for different user goals.
> For instance in **Fig.9** in revised paper, we desire to create the 3D content consistent with the prompt *''An astronaut wearing a green top hat and riding a red horse''*.
> We find that MVDream fails to create the precise result while generating *''An astronaut riding a red horse''* correctly.
> Thus the desired content can be achieved by editing *''An astronaut riding a red horse''* within one-step editing, instead of starting from *''An astronaut''*.
> Furthermore, as discussed in **Appx.A.2**, different object generating orders in Progressive3D typically result in correct 3D content consistent with the complex prompts (See in **Fig.10** in revised paper).
> We provide a general prompt decompose criterion, *i.e.,* we first generate the primary object which is desired to occupy most of the space and interact with other additional objects.
> In addition, the prompt decomposition can be achieved through LLMs in a similar way with BLIP-VQA (See in **Appx.B.2**).
> In our experiments, we generate all results step-by-step manually and add one additional object in one step for quantitative evaluation.
>
>
> > **Q7:** Report the CLIP-Score.
>
> **A7:**
> We underline that CLIP is verified that fail to measure the fine-grained correspondences between objects and binding attributes.
> For instance, we report the CLIP and our leveraged BLIP-VQA of the baseline result and our result with a specific prompt in Tab.1.
>
> **Table 1:** Quantitative comparison on sample with prompt *"An astronaut holding a red rifle and riding a green origami motorcycle and wearing a cyan chef’s hat."* shown in **Fig.1** in revised paper.
> | Method | CLIP| BLIP-VQA|
> | -------| ---------|---|
> | DreamTime | 0.340 | 0.062 |
> | + Progressive3D | 0.312 | 0.556 |
> ||
>
> We find that CLIP believes that the baseline result is more similar to the given prompts.
> However, 3D content generated by Progressive3D is significantly more accurate according to the visualization in **Fig.1** in revised paper, and BLIP-VQA shows consistent result with the visualization.
> We report the CLIP result over CSP-100 in Tab.2.
>
> **Table 2:** Quantitative comparison on metrics and user studies over CSP-100
> | Method | CLIP|
> | -------| --------------------- |
> | DreamTime | 0.289 |
> | + Progressive3D | 0.292 |
> ||
>
> --------------
>
> **Reference:**
>
> [1] Wang, Zhengyi, et al. "ProlificDreamer: High-Fidelity and Diverse Text-to-3D Generation with Variational Score Distillation." arXiv preprint arXiv:2305.16213 (2023).

---

> > ### Comment · Reviewer_S6PX · 2023-11-22
> > **Main text more than 9 pages.**
> >
> > There will be a strict upper limit of 9 pages for the main text of the submission, with unlimited additional pages for citations. Now it is more than 10 pages.

---

> > > ### Author Response · Authors · 2023-11-22
> > > **Response to Reviewer S6PX**
> > >
> > > We sincerely appreciate your invaluable advice. We now resubmit a new revised paper with a limit of 9 pages for the main text. We have moved the Preliminary in the Methods section and the discussion of attribute editing and region definitions into the Appendix. We humbly expect you can check the replies and reconsider the decision.

---

> ### Comment · Reviewer_S6PX · 2023-11-22
> **Still have some concerns.**
>
> The authors have addressed some of my concerns, including the attributes, dataset, and time cost. However, there are still some issues that remain unclear to me:
>
> - Regarding the text prompt 'An astronaut riding a red horse' in Figure 9 of the revised paper, how can you determine the position of the 3D bounding box for the green top hat?
> - Despite revisions to Figures 1 and 5, most of the 3D-generated objects still appear to be of low quality.
> - The clip-score has not been provided in the revised paper.
>
> After reviewing the revised version submitted by the authors, I have decided to increase the score to 5.

---

> ### Author Response · Authors · 2023-11-22
> **Response to Reviewer S6PX**
>
> We sincerely appreciate your invaluable advice again. Here are our detailed replies to your questions and a revised paper has been submitted. We humbly expect you can check the replies and reconsider the decision.
>
> > **Q1:** How to determine the position of the 3D bounding box in Fig.9
>
> **A1:**
> The 3D bounding box is set manually and the visualization of the 3D box is shown in **Fig.9** in the revised paper.
>
> > **Q2:** 3D objects are low-quality except for **Fig.1&5**
>
> **A2:**
> We have provided the MVDream-based high-quality qualitative comparison results in **Fig.3,7,9,10,11,15**.
> We highlight that our Progressive3D is **general** for text-to-3D methods, and **MVDream is a concurrent work (Submitted to ICLR2024)**.
> We will provide more qualitative and quantitative results on MVDream+Progressive3D in the final version.
>
>
> > **Q3:** The clip-score has not been provided in the revised paper
>
> **A3:**
> We have provided comparisons in **Fig.14** to demonstrate that the CLIP metric fails to measure the fine-grained correspondences while BLIP-VQA performs well, and we report the quantitative comparison of 4 DreamTime-based methods on CLIP metric over CSP-100 in **Tab.3** in the revised paper.

---

### Author Response · Authors · 2023-11-22
**Response to All Reviewers**

We sincerely appreciate the invaluable feedback provided by reviewers. We have carefully replied to your questions and suggestions and have submitted a revised version of the paper. We mark the modified sections in **blue**.  We humbly expect you can check the **replies & revised paper** and reconsider the decision.

---

### Author Response · Authors · 2023-11-23
**[Last one day] Reminder for the Rebuttal of Paper#2251**

Dear Reviewers,

We sincerely appreciate the invaluable feedback provided by reviewers. We have carefully replied to **all your questions and suggestions** and have submitted a **revised version** of the paper. We mark the modified sections in **blue**.
Since we only have **one day left** for the discussion, we humbly expect that you can check the **replies & revised paper** and reconsider the decision.

Thanks for your attention!

Best regards,

Authors of Paper#2251

---

### Meta-Review · Area_Chair_xyM8 · 2023-12-25

**Metareview:**

This paper presents a method for user-driven local editing for text-to-3D generation. This enables complex prompts, where text-to-3D methods struggle can struggle with composition, to be broken down progressively, providing stronger results.

The reviewers agreed that the user having to supply a bounding box is a limitation (S6PX, dBWG, EZDr), as the method requires additional inputs. However, the reviewers largely agree that the method is general and can be incorporated with existing text-to-3D methods and yields improvements (S6PX, dBWG, GY6s, EZDr). While initial reviews were split, the author rebuttal by and large addressed concerns, and 3/4 reviewers recommend acceptance. The AC agrees.

**Justification For Why Not Higher Score:**

The paper is a solid contribution to ICLR. The paper's main limitation, as pointed out by the reviewers, that additional user guidance is required to achieve better results, limits the potential of the method.

**Justification For Why Not Lower Score:**

The paper is a solid contribution to ICLR, demonstrating a method to incorporate local user guidance on complex prompts for text-to-3D generation.

---

### Decision · Program_Chairs · 2024-01-16

Accept (poster)